# A Systematic Approach to Universal Random Features in Graph Neural Networks

**Billy Joe Franks**                                                                    *billy.franks@rptu.de*
*Department of Computer Science*
*University of Kaiserslautern-Landau (RPTU)*

**Markus Anders**
*Department of Mathematics*
*TU Darmstadt*

**Marius Kloft**
*Department of Computer Science*
*University of Kaiserslautern-Landau (RPTU)*

**Pascal Schweitzer**
*Department of Mathematics*
*TU Darmstadt*

**Reviewed on OpenReview:** *https://openreview.net/forum?id=AXUtAIXOFn*

## Abstract

Universal random features (URF) are state of the art regarding practical graph neural networks that are provably universal. There is great diversity regarding terminology, methodology, benchmarks, and evaluation metrics used among existing URF. Not only does this make it increasingly difficult for practitioners to decide which technique to apply to a given problem, but it also stands in the way of systematic improvements. We propose a new comprehensive framework that captures all previous URF techniques. On the theoretical side, among other results, we formally prove that under natural conditions all instantiations of our framework are universal. The framework thus provides a new simple technique to prove universality results. On the practical side, we develop a method to systematically and automatically train URF. This in turn enables us to impartially and objectively compare all existing URF. New URF naturally emerge from our approach, and our experiments demonstrate that they improve the state of the art.

## 1 Introduction

Structured data is omnipresent in the modern world. Graphs are a fundamental data type in machine learning (ML) on structured data and used in a variety of learning algorithms (Shervashidze et al., 2011; Nickel et al., 2016; Hamilton et al., 2017), including graph neural networks (GNNs) (Zhou et al., 2020; Wu et al., 2021). GNNs, and especially message passing neural networks (MPNNs) (Gilmer et al., 2017), have found widespread applications, from drug design to friendship recommendation in social networks (Song et al., 2019; Tang et al., 2020). However, severe limitations of MPNNs have recently been proven formally. In fact, MPNNs are at most as expressive as *color refinement* (Xu et al., 2019; Morris et al., 2019). Color refinement — also known as the 1-dimensional Weisfeiler-Leman algorithm — is a simple algorithm inducing a *non*-universal similarity measure on graphs. Thus MPNNs fail to be universal function approximators, a fundamental property known for multilayer perceptrons since the 1980s (Cybenko, 1989).

Addressing this lack, universal random features (URF) (Murphy et al., 2019; Dasoulas et al., 2020; Sato et al., 2021; Abboud et al., 2021) were developed, which provably enable MPNNs to be universal function

approximators. URF enhances the nodes of the graph with random values, thus artificially breaking its intrinsic symmetries and facilitating the distinction of previously indistinguishable nodes. URF represent the state of the art in practical, efficient, and universal GNNs. Recently, numerous variations of URF have been developed (Murphy et al., 2019; Dasoulas et al., 2020; Sato et al., 2021). No systematic comparison of these URF is available and it is unclear how their trainability, generalizability, and expressivity (Raghu et al., 2017) compare to one another, because of inconsistent terminology, differing or incomplete data sets for benchmarking, and orthogonal evaluation metrics. This makes it difficult for practitioners to decide which technique to apply. It also stands in the way of systematic improvements.

**Our contributions.** We propose two systematic approaches to dealing with URF: 1) The individualization-refinement node initialization (IRNI) framework, a theoretical description that encompasses all currently known URF schemes (and even proposes new ones, see Figure 1). The foundation for our framework is graph isomorphism theory, particularly the individualization-refinement (IR) paradigm. 2) A tuning technique based on bayesian hyperparamter optimization (BHO), designed specifically to tune methods based on the IRNI framework.

Armed with these new systematic approaches, we make the following contributions:

- We demonstrate that our framework IRNI is comprehensive. In fact, we show that all previous URF techniques are captured.

- We formally prove that all instantiations of the IRNI framework that satisfy a natural compatibility condition are universal and equivariant, even ones not considered so far. Furthermore, we prove that already a very limited version of IRNI is universal on almost all graphs, including all 3-connected planar ones. We also quantify the amount of randomness required to ensure universality of all but an exponentially small fraction of all graphs.

- We apply and systematically test "ensembling over randomness" on all URF. This is a particular form of ensembling that previously has been applied in some but not all URF (Murphy et al., 2019; Dasoulas et al., 2020). Our crucial new insight is that ensembling over randomness significantly improves the performance of all URF, even in cases in which it had not been considered previously.

- The IRNI framework also suggests a new most natural URF directly related to practical graph isomorphism solvers.

- We compare all previous URF methods using the same BHO tuning approach. This leads to multiple new state-of-the-art performances. In particular, all previous URF methods are improved upon. Additionally, we discover that even though random node initialization (RNI) was reported to be harder to train than other URF (Abboud et al., 2021), our tuning approach resolves this issue.

In our evaluation, and somewhat surprisingly, we find that currently, there seems to be no clear best strategy in practice. There are strengths and weaknesses to all considered URF. However, this marks the first direct comparison of all the different URF and thus enables a more informed choice of which method to use in practice.

Our hope is that our systematic approach also eases future development of URF, in particular for developing and evaluating new algorithmic ideas.

**Individualization refinement in a nutshell:** The individualization-refinement framework is a general technique to devise algorithms for tasks revolving around symmetry. Specifically, it can be used to compute isomorphisms, automorphisms (or symmetries), and canonical forms of graphs or other combinatorial objects. The idea is first to use efficient subroutines to distinguish vertices according to simple structural properties preserved by symmetries. An example, which in fact, is similar to what is computed by color refinement (also known as the 1-dimensional Weisfeiler-Leman algorithm), iteratively collects information on the degree of neighbors of a vertex and the degrees of the neighbors of the neighbors and so on. These efficient subroutines must be isomorphism invariant and refine the partition of vertices into indistinguishable parts (and are therefore called refinements).

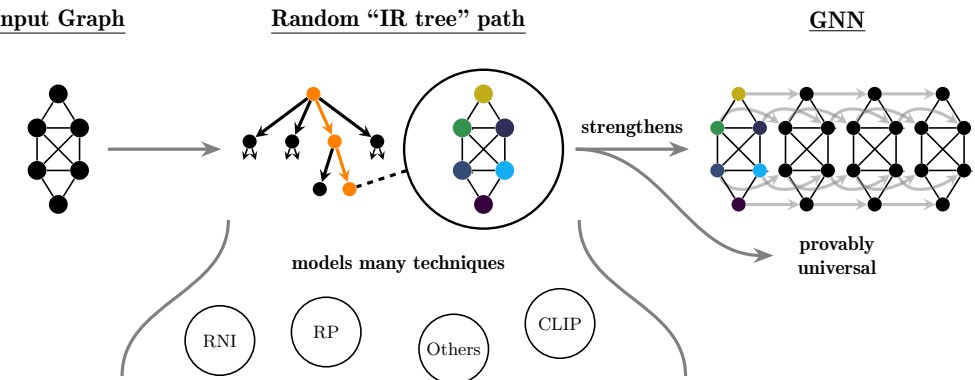

Figure 1: Summary of our new approach: we describe a framework based on "random IR tree paths", called IRNI. Through its parameters, the model is able to exactly model many previous techniques. We prove that any IRNI-enhanced GNN is universal.

Once a state is reached where refinements no longer yield new information, an IR algorithm artificially distinguishes a vertex $v$ with a so-called individualization. This individualization breaks apparent symmetry and allows us to recurse by applying another refinement step, and so on. Since the individualization is artificial, to preserve isomorphism-invariance we also have to perform the individualization in a backtracking manner with all vertices which have not been distinguished from the vertex $v$ at this point. The backtracking process finishes once all vertices have been distinguished, making symmetry computations trivial. We refer to Section 4 for a more in-depth explanation of the IR machinery necessary for this work.

**Why individualization refinement.** We explain why the IR-framework is a natural choice for the development of efficient universal GNNs. (For general strengths and weaknesses beyond ML see (McKay & Piperno, 2014; Neuen & Schweitzer, 2017).) First of all, the introduction of graph isomorphism techniques in the context of machine learning on graphs already led to two success stories, namely the efficient Weisfeiler-Leman (WL) Kernels (Shervashidze et al., 2011) based on color refinement (1-WL) and the more theoretical higher-order graph neural networks (Morris et al., 2019) based on higher-dimensional WL.

However, when it comes to practical graph isomorphism, the use of the 2-dimensional WL is already prohibitive. This is not only due to excessive running time but also due to excessive memory consumption. In the world of isomorphism testing, higher-dimensional Weisfeiler-Leman algorithms remain on the theoretical side. In truth, without fail, modern solvers are IR algorithms (McKay, 1981; Junttila & Kaski, 2011; McKay & Piperno, 2014; Anders & Schweitzer, 2021a). They only use color refinement and instead of higher-dimensional WL rather use individualizations to achieve universality. In contrast to higher-dimensional WL, the IR approach is generic, universal, *and* practical. Uncontested for more than 50 years, IR algorithms have the fastest computation times and acceptable memory consumption for graph isomorphism (McKay & Piperno, 2014). In that sense, the IRNI approach we introduce in this paper is the first time in which universal graph isomorphism techniques that are truly used in practice are transferred into a machine learning context. An important consequence of this is that ML practitioners can now readily transfer existing IR-related results to ML on graphs (Sec. 4.3).

## 2  Related Work

Graphs are a powerful means of representing semantic information. Since graphs are a very general data type and most other data types are special cases, graphs have many applications. They can be used to extend or combine other data types like text, images, or time series (Noble & Cook, 2003; Vazirgiannis et al., 2018) and there are also data sets specific to graphs. Most commonly, these data sets are related to biology or chemistry. However, computer vision, social networks, and synthetic graphs without a related field are also present (Morris et al., 2020a). Neural learning on structured data like graphs was first introduced in

(Baskin et al., 1997; Sperduti & Starita, 1997). Recently, a more specific deep learning model was pioneered for graphs, the graph neural network (GNN) (Gori et al., 2005; Scarselli et al., 2009). The GNN led to the development of a multitude of related models (Duvenaud et al., 2015; Li et al., 2016) usually referred to just as GNNs. GNNs allow for the joint training of graph feature extraction and classification, which previous models did not. Gilmer et al. (2017) gave a very general characterization of GNNs called message-passing neural networks (MPNN), which most GNN models can be characterized as. Lately, multiple concepts from other domains of deep learning have been transferred to GNNs like the attention mechanism (Velickovic et al., 2018) and hierarchical pooling (Ying et al., 2018).

Cybenko (1989) proved the first universality result for one of the earliest deep learning models, the smallest possible multilayer perceptron (MLP). This result has since then been expanded to different activation functions (Leshno et al., 1993; Barron, 1994), to width-bounded MLPs (Lu et al., 2017; Liang & Srikant, 2017), and more recently to different layered artificial neural networks like the convolutional neural network (Zhou, 2020). Analogous results had been lacking for MPNNs, which are now well-established to be non-universal (Xu et al., 2019; Morris et al., 2019; Abboud et al., 2021). Following this finding, multiple attempts were made to establish universal ML models on graph data. Murphy et al. (2019) proposed relational pooling (RP), Sato et al. (2021) proposed random node initialization (RNI), and Dasoulas et al. (2020) proposed the colored local iterative procedure (CLIP), all of which provide universality to MPNNs (Abboud et al., 2021) and are closely related to one another as methods using universal random features (URF). Morris et al. (2019; 2020b) proposed k-GNNs based on the k-dimensional Weisfeiler-Leman algorithm and expanded on it by proposing the $\delta$-k-GNN a local approximation variant of the k-GNN. Maron et al. (2019a;b) propose provably powerful graph networks, which are 2-WL powerful, and invariant graph networks, which are proven to be universal, see Morris et al. (2021) for further pointers.

In this paper, we only consider methods that are scalable, practical, and universal at the same time. For a comparison between universal and non-universal approaches we refer to existing literature (Dasoulas et al., 2020; Morris et al., 2019; 2020b; Maron et al., 2019a). As expected non-universal approaches can outperform universal approaches in tasks that do not require high expressivity. However, non-universal approaches fail to achieve high performance on tasks that require high expressivity. Therefore, only methods employing random features that grant universality (Murphy et al., 2019; Dasoulas et al., 2020; Sato et al., 2021), or universal random features (URF), are considered. Some readers might wonder if typical graph data augmentation schemes should also be considered here. Examples would include deleting/adding edges/nodes. While these changes to graphs can induce the capacity to distinguish graphs from one another, we expected that universality of these methods is hard to prove. For deletions, this is essentially due to the various reconstruction conjectures, which so far have no proof. It is currently unknown if the set of all graphs with one node/edge deleted can be used to reconstruct the original graph, which would be required for these operations to be universal. As for node addition, the IRNI framework subsumes this operation as coloring a set of nodes is equivalent to node addition. Lastly, edge addition can be viewed as removing edges from the inverted graph and therefore has the same issues as edge deletion. For these reasons, we do not consider the vast number of data augmentations that are not provably universal (Puny et al., 2020; Papp et al., 2021; Ding et al., 2022).

## 3 Background

Here we cover existing background necessary to understand the IRNI framework. Specifically we define graphs as well as how they are used in ML followed by coloring of graphs. We then cover color refinement a fundamental backbone of individualization refinement, the practical algorithm the IRNI framework is based on. This is followed by a definition of GIN networks the specific MPNN model we use. Lasty, we cover the specifics of RP, RNI, and CLIP, which are the related methods that can be described in the IRNI framework and that we compare later on.

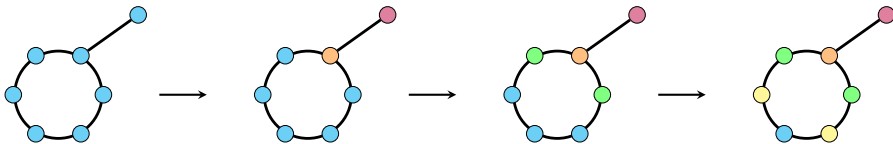

Figure 2: A run of naive refinements on a graph. The sequence of naive refinements ends in the coarsest equitable coloring, which can not be refined further using the naive refinement. Note how each coloring is finer than the previous coloring.

### 3.1 Graphs and Colorings

We consider undirected, finite graphs $G = (V, E)$ which consist of a set of vertices $V \subseteq \mathbb{N}$ and a set of edges $E \subseteq V^2$, where $E$ is symmetric. From this point onward, let $n := |V|$ and $V = \{1, \ldots, n\}$. Additionally, we let $\mathcal{G}$ denote the set of all graphs, while $\mathcal{G}_n$ denotes the set of all graphs on $n$ vertices.

In ML contexts, graphs typically carry a node representation in $\mathbb{R}^d$, which we denote by $X = \{\boldsymbol{x}_1, \ldots, \boldsymbol{x}_n\}$. IR-tools require these node representations to be discrete. In other contexts, discretization can be difficult and techniques are being actively researched (Morris et al., 2016). However, the discretization is not critical for our purpose since we only require this encoding to compute a tuple of nodes $(w_1, w_2, \ldots)$ from an IR algorithm. After that, our approach continues on the original node representation. Let $enc : \mathbb{R}^d \times \mathcal{G} \to \mathbb{N}$ be an arbitrary isomorphism-invariant encoding of the node representations. In practice, it is best to choose an encoding for which $enc(\boldsymbol{x}_v, G) = enc(\boldsymbol{x}_w, G)$ if and only if $\boldsymbol{x}_v = \boldsymbol{x}_w$, however, this is not a requirement for any of the results we present. Elaborating further, if $enc(\boldsymbol{x}_v, G) = enc(\boldsymbol{x}_w, G)$ even though $\boldsymbol{x}_v \neq \boldsymbol{x}_w$ then more nodes might be "individualized" than are strictly necessary, still resulting in a universal and equivariant method as described in Sec. 4.

A (node) *coloring* is a surjective map $\pi : V \to \{1, \ldots, k\}$. We interpret the node representations as colors using $enc$, i.e. $\pi(i) := enc(\boldsymbol{x}_i, G)$. We call $\pi^{-1}(i) \subseteq V$ the $i$-th cell for $i \in \{1, \ldots, k\}$. If $|\pi(V)| = n$ then $\pi$ is *discrete*. This means every node has a unique color in $\pi$. Note that in the following, we always use "discrete" in this sense. Furthermore, we say a coloring $\pi$ is *finer* than $\pi'$ if $\pi(v) = \pi(v') \implies \pi'(v) = \pi'(v')$ holds for every $v, v' \in V$. We may also say $\pi'$ is *coarser* than $\pi$. See Figure 2, where each consecutive coloring of the graph is finer than the previous one.

We should remark that the IR machinery generalizes to directed and edge-colored graphs. One example of achieving this is by subdividing each edge with additional colored vertices, where the additional vertices model the edge color or edge direction (see McKay & Piperno for more details). Alternatively, said features can also be added to IR algorithms directly (Piperno, 2018).

We denote by $N_G(v)$ the neighborhood of node $v$ in graph $G$.

### 3.2 Color Refinement

We now define color refinement, which all practical graph isomorphism solvers use. A coloring $\pi$ is *equitable* if for every pair of (not necessarily distinct) colors $i, j$, the number of $j$-colored neighbors is the same for all $i$-colored vertices. Equitable colorings are precisely the colorings for which color refinement cannot be employed to distinguish nodes further. For a colored graph $(G, \pi)$ there is (up to renaming of colors) a unique coarsest equitable coloring finer than $\pi$ (McKay, 1981). This is precisely the coloring computed by color refinement.

A more algorithmic way to describe the refinement is to define for a colored graph $(G, \pi)$ the naively refined graph $(G, \pi^r)$ where $\pi^r(v) := (\pi(v), \{\!\!\{\pi(v') \mid v' \in N_G(v)\}\!\!\})$. The naive refinement $r$ is applied exhaustively, i.e., until vertices cannot be partitioned further. The result is precisely the coarsest equitable coloring. Figure 2 illustrates the color refinement process.

Note that color refinement, and hence coarsest equitable colorings, are strongly related to MPNNs, as is explained below.

### 3.3 Message Passing Neural Networks

Formally a message passing update can be formulated as:

$$\boldsymbol{x}_{v,t+1} := \text{combine}(\boldsymbol{x}_{v,t}, \text{aggregate}(\{\!\!\{\boldsymbol{x}_{w,t} | w \in N_G(v)\}\!\!\})), \tag{1}$$

where $G$ is a graph and $\boldsymbol{x}_{v,t}$ is the vector representation of node $v$ at time $t$. In this context, time t is usually interpreted as the layer and the aggregate function is typically required to be invariant under isomorphisms. We want to point out the similarity between the refinement $r$ in color refinement and the message passing update (Equation 1). MPNNs are just computing on the colors computed by color refinement, which is why MPNNs are at most as powerful as color refinement in distinguishing graphs. We refer to Xu et al. (2019) and Morris et al. (2019) for a formal comparison. Common instances of MPNNs are graph convolutional networks (Duvenaud et al., 2015), graph attention networks (Velickovic et al., 2018), and graph isomorphism networks (GIN) (Xu et al., 2019). We use GIN throughout this paper, arguably the most efficient of these models (Dasoulas et al., 2020).

A GIN is characterized by being simple and yet as powerful as the color refinement algorithm. Given an arbitrary multi-layer perceptron $\text{MLP}^{(k)}$ in layer $k$, $\epsilon^{(k)}$ a learnable parameter of layer $k$, and $h_v^{(0)}$ the input representation of node $v$. GIN updates its node representations $h_v^{(k)}$ in layer $k$ as follows:

$$h_v^{(k)} := \text{MLP}^{(k)} \left( \left(1 + \epsilon^{(k)}\right) h_v^{(k-1)} + \sum_{u \in \mathcal{N}_G(v)} h_u^{(k-1)} \right). \tag{2}$$

### 3.4 Random Features

Intuitively, using random features is a process that introduces randomized information into the input before processing it using machine learning techniques. For example, one might delete a random node or attach random numbers to feature vectors. We do not require a more formal notion since, in this paper, we are only interested in universal random features (URF). We use URF to refer to methods that provably enable their universality if used in conjunction with MPNNs. We next summarize all existing URF techniques.

RP attaches a random permutation to the graph by attaching to each node a one-hot encoding of its image. To understand RNI, let $G$ be a graph with node representations $\{\boldsymbol{x}_1, \ldots, \boldsymbol{x}_n\}$ and $d \in \mathbb{N}$ a constant, random node initalization (RNI) concatenates $d$ features sampled from a random distribution $\mathcal{X}$ to each node $\forall v \in \{1, \ldots, n\} : \boldsymbol{x}_v \leftarrow \text{concatenate}(\boldsymbol{x}_v, r_1, \ldots, r_d), r_1 \ldots, r_d \sim \mathcal{X}$. CLIP first apply color refinement and then individualize each node of each color class by assigning a one-hot encoding of a natural number to it. If $C$ is the set of all nodes of one color class, then each node $v \in C$ is randomly assigned a unique number in $\{0, \ldots, |C| - 1\}$, which is then one-hot encoded and concatenated onto its node representation $\boldsymbol{x}_v$.

## 4 Random Features from Individualization Refinement

First, we describe individualization-refinement trees followed by the individualization-refinement node initialization (IRNI) framework. Then, we demonstrate that RNI, CLIP, and RP can be expressed as a manifestation of this framework. Lastly, we give several theoretical insights into the framework and prove a general universality for these methods under a natural compatibility constraint.

### 4.1 Individualization Refinement Trees

Individualization refinement (IR) trees are the backtracking trees of practical graph isomorphism algorithms. We use randomly sampled leaves of these trees as random features. These leaves correspond to sequences of nodes $(w_1, \ldots, w_k)$ of the graph, which we translate into features of the MPNN. One central property of the sequence is that distinguishing its nodes from other nodes in the graph and applying color refinement yields a discrete coloring.

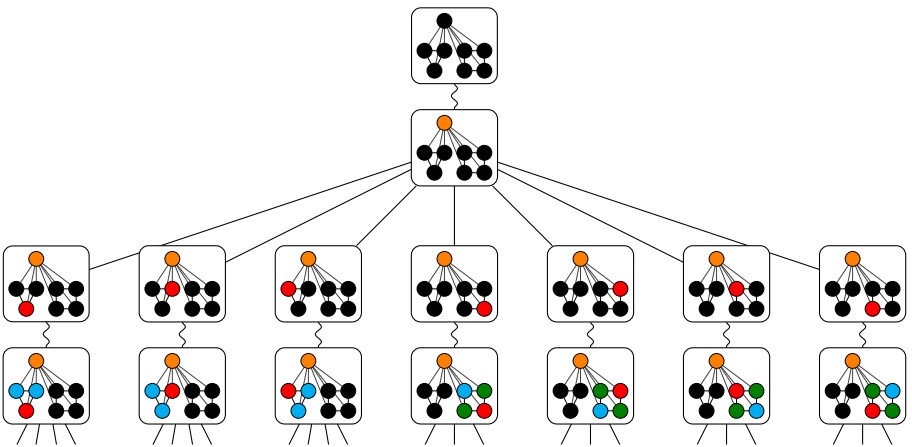

Figure 3: In IR, refinement and individualization are alternatingly applied. This continues until the coloring of the graph becomes discrete. Two nodes connected by a squiggly line are considered one node of the IR tree: they illustrate the coloring of the graph before and after color refinement.

In the following, we describe all the necessary ingredients of IR for the purposes of this paper. The IR paradigm is a complex machinery refined over many decades into sophisticated software libraries. We refer to McKay & Piperno (2014) and Anders & Schweitzer (2021a) for an exhaustive description.

**Refinement.** The most crucial subroutine of an IR algorithm is the *refinement* Ref : $G \times \Pi \times V^* \to \Pi$, where $\Pi$ denotes the set of all vertex colorings of $G$ and $V^*$ denotes a string of vertices ($^*$ is the Kleene star). A refinement must satisfy two properties: it must be invariant under isomorphism and individualize vertices in $\nu \in V^*$, i.e., let $\pi' = \text{Ref}(G, \pi, \nu)$, then for all $v \in \nu$ it holds that $\pi'^{-1}(\pi'(v)) = \{v\}$. Our definition of refinement is slightly more general compared to McKay & Piperno (2014), which leads to a slight technicality that we discuss in the appendix.

In practice, IR tools use color refinement as their refinement (see Sec. 3.2). We denote color refinement as $\text{CR}(G, \pi, \epsilon)$, where $\epsilon$ denotes the empty sequence (explained further below).

**Individualization.** IR algorithms make use of *individualization*, a process that artificially forces a node into its own color class, distinguishing it. To record which vertices have been individualized we use a sequence $\nu = (v_1, \ldots, v_k) \in V^*$. We modify color refinement so that $\text{CR}(G, \pi, \nu)$ is the unique coarsest equitable coloring finer than $\pi$ in which every node in $\nu$ is a singleton with its own artificial color. Artificial distinctions caused by individualizations are thus taken into account.

**Cell selector.** In a backtracking fashion, the goal of an IR algorithm is to reach a discrete coloring using color refinement and individualization. For this, color refinement is first applied. If this does not yield a discrete coloring, individualization is applied, branching over all vertices in one non-singleton cell. The task of the *cell selector* is to (isomorphism-invariantly) pick the non-singleton cell. Figure 3 illustrates this process. While many choices within certain restrictions are possible, one example that we will also use later on is the selector that always chooses the first, largest non-singleton cell of $\pi$. We use the notation Sel to refer to a cell selector.

**IR trees.** We first give a formal definition of the IR tree $\Gamma_{\text{Ref,Sel}}(G, \pi)$, followed by a more intuitive explanation. Nodes of $\Gamma_{\text{Ref,Sel}}(G, \pi)$ are sequences of vertices of $G$ and the root is the empty sequence $\epsilon = ()$. If $\nu = (v_1, \ldots, v_k)$ is a node in $\Gamma_{\text{Ref,Sel}}(G, \pi)$ and $C = \text{Sel}(G, \text{Ref}(G, \pi, \nu))$ is the selected cell, then the set of children of $\nu$ is $\{(v_1, \ldots, v_k, v) \mid v \in C\}$, i.e., all extensions of $\nu$ by one node $v$ of the selected cell $C$. The root represents the graph (with no individualizations) after refinement (see Figure 3). A node $\nu$ represents the graph after all nodes in $\nu$ have been individualized followed by refinement (see Figure 3). A root-to-$\nu$

walk of the tree is naturally identified with a sequence of individualizations in the graph: in each step $i$ of the walk, one more node $v_i$ belonging to a non-trivial color class $C$ is individualized (followed by refinement). The sequence of individualizations $(v_1, \ldots, v_k)$ uniquely determines the node of the IR tree in which this walk ends. This is why we identify the name of the node with the sequence of individualizations necessary to reach the node: the sequence of individualizations necessary to reach $\nu$ is $(v_1, \ldots, v_k) = \nu$. $\Gamma_{\text{Ref,Sel}}(G, \pi, \nu)$ denotes the subtree of $\Gamma_{\text{Ref,Sel}}(G, \pi)$ rooted in $\nu$. We remark that the notation $(G, \pi)^\varphi$ simply means we apply $\varphi$ to $G$ and $\pi$, as noted in the lemma below. Isomorphism invariance of the IR tree follows from isomorphism invariance of Sel and Ref:

**Lemma 1** (McKay & Piperno (2014)). *Let $\varphi \colon V \to V$ denote an* automorphism *of $(G, \pi)$, i.e., $(G, \pi)^\varphi = (\varphi(V), \varphi(E), \varphi(\pi)) = (V, E, \pi) = (G, \pi)$. Let $\text{Aut}(G, \pi)$ denote all automorphisms of $(G, \pi)$. Then, if $\nu$ is a node of $\Gamma_{\text{Ref,Sel}}(G, \pi)$ and $\varphi \in \text{Aut}(G, \pi)$, then $\nu^\varphi$ is a node of $\Gamma_{\text{Ref,Sel}}(G, \pi)$ and $\Gamma_{\text{Ref,Sel}}(G, \pi, \nu)^\varphi = \Gamma_{\text{Ref,Sel}}(G, \pi, \nu^\varphi)$.*

Generally, we refer to a process or object, as *isomorphism-invariant*, whenever it produces the same result for isomorphic inputs.

We want to remark again that leaves of an IR tree correspond to discrete colorings of a graph. In fact, the set of all leaves of an IR tree forms a complete isomorphism invariant: two isomorphic graphs will have the same set of leaves, whereas two non-isomorphic graphs are guaranteed to have a distinct set of leaves (see Lemma 4 of McKay & Piperno (2014)).

**Random IR walks.** There are various ways to traverse and use IR trees. Traditionally, solvers (e.g., NAUTY) solely used deterministic strategies, such as depth-first traversal (McKay & Piperno, 2014). Only recently competitive strategies based solely on random traversal, i.e., DEJAVU (Anders & Schweitzer, 2021a), have emerged (see Section 4.3). We make use of this recent development, by using *random root-to-leaf walks* of the IR tree $\Gamma_{\text{Ref,Sel}}(G, \pi)$. We begin such a walk in the root node of $\Gamma_{\text{Ref,Sel}}(G, \pi)$. We repeatedly choose uniformly at random a child of the current node until we reach a leaf $\nu$ of the tree. Then, we return the leaf $\nu = (w_1, \ldots, w_k)$. A crucial property is that since $\Gamma_{\text{Ref,Sel}}(G, \pi)$ is isomorphism-invariant (Lemma 1), random walks of $\Gamma_{\text{Ref,Sel}}(G, \pi)$ are isomorphism-invariant as well:

**Lemma 2** (Anders & Schweitzer (2021a)). *As a random variable, the graph colored with the coloring of the leaf resulting from a random IR walk is isomorphism-invariant.*

Lemma 2 is also true when restricting random walks to prefixes of a certain length $d$. We stress that random IR walks are conceptually unrelated to random walks in the graph itself considered elsewhere (Nikolentzos & Vazirgiannis, 2020). Our next step is to insert the sequence of nodes defined by random IR walks into MPNNs.

## 4.2 Individualization Refinement Node Initalization

Let $G$ be a graph with node representations $\{\boldsymbol{x}_1, \ldots, \boldsymbol{x}_n\}$ and $d \in \mathbb{N}$ a constant. Individualization-refinement node initalization (IRNI) computes a random IR walk $w = (w_1, \ldots, w_k)$ in $\Gamma_{\text{Ref,Sel}}(G, \pi)$, where $\pi(i) := enc(\boldsymbol{x}_i, G)$. If $k \geq d$, we take a prefix of length $d$, i.e., $(w_1, \ldots, w_d)$, providing $d$ nodes to be individualized. IRNI then concatenates $d$ features that are either 0 or 1 depending on this prefix: We set $\forall v \in \{1, \ldots, n\}$ : $\boldsymbol{x}_v \leftarrow concatenate(\boldsymbol{x}_v, \mathbb{1}_{w_1=v}, \ldots, \mathbb{1}_{w_d=v})$, which means that the $j$-th feature of node $v$ is set to 1 if $v$ is the $j$-th node that was individualized (i.e., $w_j = v$) and 0 otherwise. This guarantees that node $v$ is individualized if and only if it appears in the prefix. If $k < d$, then we simply "fill up" the walk with nodes in an isomorphism-invariant manner using the discrete coloring $\text{Ref}(G, \pi, (w_1, \ldots, w_k))$: we add nodes in order of their color, first the node with the smallest color, then with the second smallest color, and so forth. We abbreviate IRNI with constant $d$ as $d$-IRNI.

Due to the dependence on an underlying IR-tree, both the refinement Ref and cell selector Sel are natural hyperparameters of IRNI. Unless stated otherwise, we assume that an arbitrary refinement and an arbitrary cell selector are used. If we want to state a specific refinement or cell selector, we do so using $d$-IRNI(Ref) and $d$-IRNI(Ref, Sel), respectively.

IRNI depends on the random walk in the IR tree and we will prove its universality in Theorem 4. Thus, IRNI is a URF (analogous to RNI, CLIP, and RP). This justifies ensembling over this randomness, which we will refer to as ensembling over randomness (EoR). Specifically, we average the predictions of an MPNN over some URF.

We now define some specific instances of IRNI by applying different refinements. Let the trivial refinement $\mathrm{TR}(G, \pi, \nu)$ only individualize the vertices $\nu$ in $\pi$, followed by no further refinement. A random walk of $\Gamma_{\mathrm{TR}}(G, \pi)$ thus picks a random permutation of vertices that respects only the initial color classes of $\pi$ (i.e., the first vertex will always be of the first selected color of $\pi$, and so forth). In case of uncolored graphs $(G, \pi)$ where $\pi$ is the trivial coloring, random walks truly only become random permutations of vertices of $G$. In this case, it follows that for $G \in \mathcal{G}_n$ $n$-IRNI(TR) is equivalent to RP. However, we can enforce this even for colored graphs by actively ignoring the colors of $\pi$, resulting in what we call the *oblivious refinement* $\mathrm{OR}(G, \pi, \nu) := \mathrm{TR}(G, V(G) \mapsto 1, \nu)$. In this case, random walks are always random permutations of vertices of $G$. Hence, it follows that for $G \in \mathcal{G}_n$ $n$-IRNI(OR) is RP. We remark that the only difference between RNI and RP is in the encoding of the individualizations.

Based on TR and CR, we define $\mathrm{CTR}(G, \pi, \nu) := \mathrm{TR}(G, \mathrm{CR}(G, \pi, \epsilon), \nu)$. Note that CTR applies color refinement to the graph, followed by trivial refinement of nodes in the resulting color classes. By definition, for $G \in \mathcal{G}_n$ $n$-IRNI(CTR) is thus an alternative description of CLIP.

### 4.3 IR algorithms and IRNI

We now discuss the relationship between IR algorithms and MPNNs using IRNI. First, we remark that in terms of solving graph isomorphism, the use of repeated random IR walks has recently been proven to be a near-optimal traversal strategy of IR trees (Anders & Schweitzer, 2021b), where near-optimal refers to a logarithmic gap between the lower and upper bound. This is in contrast to deterministic traversal strategies such as depth-first search or breadth-first search, which have a quadratic overhead (Anders & Schweitzer, 2021b). URF are thus closely related to these optimal strategies. The ensembling defined in the previous section even seems to mimic the way the aforementioned near-optimal IR algorithm operates (Anders & Schweitzer, 2021b). In fact, the currently fastest practical graph isomorphism algorithm DEJAVU (Anders & Schweitzer, 2021a) uses essentially the same strategy. Moreover, the use of random IR walks has additional inherent benefits, such as "implicit automorphism pruning", i.e., the automatic exploitation of symmetry in the input (Anders & Schweitzer, 2021a). This translates to MPNNs with IRNI, in that if individualizations across multiple random walks are made on nodes that are symmetrical to each other, this does not introduce any additional randomness: due to their isomorphism-invariance (or equivariance), symmetrical nodes are indistinguishable by MPNNs by design.

Previously, we discussed that a crucial property of MPNNs is that their result is isomorphism-invariant, i.e., it only depends on the isomorphism type. This is not true in the deterministic sense for IRNI. However, because of Lemma 2, the result only depends on the isomorphism type and the randomness.

**Lemma 3.** *Let* Ref *be any refinement. Let $f$ be the function computed by an MPNN with $d$-IRNI(Ref), mapping a graph $G$ and a random seed $s \in \Omega$ from the sample space of random IR walks $\Omega$ to a value $f(G, s) \in \mathbb{R}$. Then for every permutation $\varphi$, we have that the random variables $s \mapsto f(G, s)$ and $s \mapsto f(\varphi(G), s)$ have the same distribution.*

*Proof.* The result follows directly from the isomorphism-invariance of MPNNs and Lemma 2. Nevertheless, we give a more extensive proof here.

First, we recall that MPNNs are isomorphism-invariant (or equivariant): for every permutation $\varphi$, by definition, a MPNN produces the same result for $G$ as for $\varphi(G)$. Hence, we only need to proof that the augmentations made through random IR walks satisfy the claim.

We note that the IR tree of $G$ is $\Gamma_{\mathrm{Ref}}(G)$ and the IR tree of $\varphi(G)$ is $\Gamma_{\mathrm{Ref}}(\varphi(G)) = \varphi(\Gamma_{\mathrm{Ref}}(G))$ (see Lemma 1 in McKay & Piperno (2014)). Intuitively, this means that ignoring the permutation $\varphi$ we are drawing randomly from the same distribution.

Hence, randomly sampling walks from these trees will result in the same nodes, except for applying the permutation $\varphi$: if $\nu \in \Gamma_{\text{Ref}}(G)$, then $\varphi(\nu) \in \Gamma_{\text{Ref}}(\varphi(G))$. We remark that $G$ individualized with $\nu$ is isomorphic to $\varphi(G)$ individualized with $\varphi(\nu)$. This corresponds to the augmentation made to the MPNN in IRNI. Hence, overall, it follows that the MPNN must give the same result for $G$ augmented with $\nu$, as well as $\varphi(G)$ augmented with $\varphi(\nu)$. $\qquad\square$

We now provide a universality theorem for IRNI, in a similar fashion as Abboud et al. (2021) does for RNI. Let us first recall a definition of Abboud et al. (2021): let $\mathcal{G}_n$ denote the class of all $n$-vertex graphs and $f \colon \mathcal{G}_n \to \mathbb{R}$. We say that some randomized function $\mathcal{X}$ that associates with a graph $G \in \mathcal{G}_n$ a random variable $\mathcal{X}(G)$ is an $(\epsilon, \delta)$-approximation of $f$ if for all $G \in \mathcal{G}_n$ it holds that $\Pr(|f(G) - \mathcal{X}(G)| \leq \epsilon) \geq 1 - \delta$.

The theorem proven by Abboud et al. (2021) is based on the fact that RNI fully individualizes a graph with high probability, and all fully individualized representations of a graph together constitute a complete isomorphism invariant. The crucial insight we exploit is that IR trees constitute a complete isomorphism invariant as well, specifically, even the set of all leaves of an IR tree suffices. Since the power of MPNNs is limited by color refinement, the only additional requirement needed is that the refinement used for random walks must be at most as powerful as color refinement.

**Theorem 4.** *Let* Ref *be a refinement that computes colorings coarser or equal to* CR, *i.e., for any graph* $G = (V, E)$, *coloring* $\pi$, *and* $\nu \in V^*$, $\text{Ref}(G, \pi, \nu)$ *is coarser or equal to* $\text{CR}(G, \pi, \nu)$. *Let* $n \geq 1$ *and let* $f \colon \mathcal{G}_n \to \mathbb{R}$ *be invariant. Then, for all* $\epsilon, \delta > 0$, *there is an MPNN with* $(n-1)$-IRNI(Ref) *that* $(\epsilon, \delta)$-*approximates* $f$.

*Proof.* We prove the theorem using a combination of Theorem 2 from (Morris et al., 2019), the universality result of RNIs given in (Abboud et al., 2021), and the basic definition of IR trees. Since graphs have $n$ nodes all possible random IR walks considered by $(n-1)$-IRNI(Ref) are random IR walks ending in a leaf node of $\Gamma_{\text{Ref}}(G, \pi)$ (see Section 4.1). If we were to individualize the sequence of nodes $(w_1, \ldots, w_k)$ corresponding to a leaf and apply the refinement Ref, the coloring of the entire graph would become discrete. By assumption, color refinement always produces colorings finer or equal to Ref, so applying color refinement also produces a discrete coloring.

By the definition of $(n-1)$-IRNI(Ref), the nodes contained in $\{w_1, \ldots, w_k\}$ all have distinct features not shared by any of the other nodes in the graph. This means that the nodes in $\{w_1, \ldots, w_k\}$ are indeed initially individualized in the MPNN. Now, it is well-known that Theorem 2 of Morris et al. (2019) (see also (Xu et al., 2019)) shows that there is an MPNN that produces the same partitioning of colors that color refinement would, i.e., in our case, yields a discrete partitioning of vertices. In other words, we can assume that the graph is individualized.

This suffices to apply the universality result of Abboud et al. (2021) (see Lemma A2, Lemma A3, and Lemma A4 in (Abboud et al., 2021), which build upon (Barceló et al., 2020)), which solely depends on individualizing the graph. In particular, the proof of Abboud et al. (2021) builds a $\mathcal{C}^2$ sentence which identifies discretely colored graphs (Lemma A3). In turn, a disjunction identifying any possible discretely colored graph for a given graph is constructed (Lemma A4). Since the set of leaves of an IR tree are a complete isomorphism invariant, it suffices to build this disjunction over only those discretely colored graphs that correspond to a leaf in the IR tree. $\qquad\square$

The hyperparameters of IR open up more opportunities to transfer results into the realm of MPNNs. We give one such example. We argue that with a specific cell selector, 3-connected planar graphs can be detected with 4-IRNI(CR).

**Theorem 5.** *Let* $\mathcal{P}_n$ *denote the class of 3-connected planar graphs. Let* $n \geq 1$ *and let* $f \colon \mathcal{P}_n \to \mathbb{R}$ *be invariant. Then, for all* $\epsilon, \delta > 0$, *there is a cell selector* Sel *(which does not depend on $n$) and an MPNN with* 4-IRNI(CR, Sel) *that* $(\epsilon, \delta)$-*approximates* $f$.

*Proof.* First of all, we argue that individualizing a node of degree 5 and three of its neighbors surely suffices to make the graph discrete: this follows from Lemma 22 of Kiefer et al. (2017), which proves that individualizing 3 vertices on any common face followed by color refinement suffices to make the coloring discrete. Note that

| Method | PROTEINS | MUTAG | NCI1 | TRI | TRIX | EXP | CEXP | CSL |
|---|---|---|---|---|---|---|---|---|
| None | $0.68 \pm 0.06$ | $0.89 \pm 0.06$ | $0.81 \pm 0.02$ | $0.50 \pm 0.00$ | $0.50 \pm 0.00$ | $0.50 \pm 0.01$ | $0.74 \pm 0.02$ | $0.50 \pm 0.00$ |
| RNI | $0.66 \pm 0.02$ | $\mathbf{0.89 \pm 0.04}$ | $\mathbf{0.81 \pm 0.01}$ | $\mathbf{0.99 \pm 0.01}$ | $\mathbf{0.99 \pm 0.00}$ | $\mathbf{0.97 \pm 0.03}$ | $0.95 \pm 0.10$ | $0.85 \pm 0.06$ |
| CLIP | $0.65 \pm 0.05$ | $0.85 \pm 0.09$ | $0.81 \pm 0.01$ | $\mathbf{0.99 \pm 0.00}$ | $0.81 \pm 0.05$ | $\mathbf{0.99 \pm 0.04}$ | $\mathbf{0.99 \pm 0.02}$ | $\mathbf{1.00 \pm 0.01}$ |
| RP | $\mathbf{0.74 \pm 0.04}$ | $0.86 \pm 0.07$ | $0.81 \pm 0.01$ | $\mathbf{0.99 \pm 0.00}$ | $0.82 \pm 0.03$ | $0.96 \pm 0.02$ | $0.97 \pm 0.02$ | $\mathbf{1.00 \pm 0.00}$ |
| IRNI(CR) | $\mathbf{0.75 \pm 0.04}$ | $0.85 \pm 0.05$ | $\mathbf{0.82 \pm 0.02}$ | $\mathbf{0.99 \pm 0.01}$ | $0.73 \pm 0.04$ | $\mathbf{0.99 \pm 0.04}$ | $0.95 \pm 0.14$ | $\mathbf{1.00 \pm 0.00}$ |
| $\text{RNI}_{EoR}$ | $0.69 \pm 0.05$ | $\mathbf{0.94 \pm 0.03}$ | $0.85 \pm 0.02$ | $\mathbf{1.00 \pm 0.00}$ | $\mathbf{1.00 \pm 0.00}$ | $0.99 \pm 0.01$ | $0.98 \pm 0.05$ | $0.93 \pm 0.06$ |
| $\text{CLIP}_{EoR}$ | $0.67 \pm 0.03$ | $0.92 \pm 0.05$ | $0.82 \pm 0.02$ | $\mathbf{1.00 \pm 0.00}$ | $0.95 \pm 0.05$ | $\mathbf{1.00 \pm 0.00}$ | $0.97 \pm 0.08$ | $\mathbf{1.00 \pm 0.00}$ |
| $\text{RP}_{EoR}$ | $\mathbf{0.78 \pm 0.04}$ | $0.84 \pm 0.12$ | $\mathbf{0.87 \pm 0.02}$ | $\mathbf{1.00 \pm 0.00}$ | $0.95 \pm 0.05$ | $\mathbf{1.00 \pm 0.00}$ | $\mathbf{1.00 \pm 0.00}$ | $\mathbf{1.00 \pm 0.01}$ |
| $\text{IRNI(CR)}_{EoR}$ | $0.74 \pm 0.04$ | $0.87 \pm 0.08$ | $0.82 \pm 0.02$ | $\mathbf{1.00 \pm 0.00}$ | $0.94 \pm 0.05$ | $0.99 \pm 0.02$ | $0.97 \pm 0.06$ | $0.99 \pm 0.02$ |
| $\text{SOTA}_{URF}$ | $0.81 \pm 0.03$ | $0.95 \pm 0.03$ | $0.88 \pm 0.01$ | $0.91 \pm \text{NA}$ | $0.93 \pm \text{NA}$ | NA | NA | NA |
| $\text{SOTA}^*_{URF}$ | $0.77 \pm 0.04$ | $0.94 \pm 0.04$ | NA | NA | NA | $0.98 \pm 0.02$ | NA | $0.91 \pm 0.07$ |

Table 1: The AUROC of a GIN network with RNI, CLIP, RP, IRNI(CR), and without any (None) of these on selected data sets. $_{EoR}$ indicates the use of ensembling over randomness. Bold entries indicate statistically significant best values, for this EoR is treated as separate from no EoR. $\text{SOTA}_{URF}$ indicates the state-of-the-art AUROC. $\text{SOTA}^*_{URF}$ indicates the state-of-the-art accuracies, note that these are not directly comparable. Specifically, $\text{SOTA}_{URF}$ and $\text{SOTA}^*_{URF}$ refer to the results from the previous evaluations in (Murphy et al., 2019; Sato et al., 2021; Dasoulas et al., 2020; Abboud et al., 2021), from which we always chose the best value.

individualizing a node of degree 5 and three of its neighbors surely individualizes three vertices on a common face. Using the arguments from the proof of Theorem 4 again, we can see that this would indeed suffice to show the claim.

It remains to be shown that there is a cell selection strategy achieving the above. In the first step, the cell selector chooses an (isomorphism-invariant) color class consisting of degree 5 vertices. We remark that, due to Euler's formula, the average degree of a planar graph is less than 6, so such a node always exists. In the next step, we choose a non-trivial class containing only neighbors of the individualized degree 5 node $v$. We argue that unless all neighbors of $v$ have been individualized, there is a non-trivial color class consisting of neighbors of $v$. Indeed, if there is a non-trivial class containing neighbors of $v$, the class may only contain such neighbors since color refinement distinguishes neighbors of $v$ from non-neighbors of $v$. Here we use that $v$ is individualized. We repeat the step of choosing a non-singleton class of neighbors of $v$ and individualizing a node within it. If at any point no non-trivial class of neighbors exists, we are done: this means that the neighbors are fully discrete. This in turn suffices to show the claim. $\qquad\square$

More results of this kind can be shown. For example, it is known that strongly regular graphs require at most $O(\sqrt{n} \log n)$ individualizations (Babai, 1980). In fact, for all but an exponentially small fraction of graphs, $d$-IRNI(CR) with small $d$ suffices.

**Theorem 6.** *There is an absolute constant $c > 1$ such that the following holds. Let $n \geq 1$ and $d \in \mathbb{N}$, then there is a graph class $\mathcal{G}'_n$ containing all but at most a $1/c^{dn}$ fraction of all graphs for which the following holds. Let $f \colon \mathcal{G}'_n \to \mathbb{R}$ be invariant, then, for all $\epsilon, \delta > 0$, there is an MPNN with $d$-IRNI(CR) that $(\epsilon, \delta)$-approximates $f$.*

*Proof.* To prove the theorem, we use the same technique as before. We only need to observe that for most graphs, after color refinement is applied, $d$ arbitrary individualizations in non-singleton cells cause discretization of the graph. This, however, is a classic theorem by Babai & Kucera (1979, Theorem 4.1) showing the fraction of graphs for which this fails is at most $1/c^{dn}$. $\qquad\square$

## 5  Experiments

We compare the URF schemes RNI, CLIP, RP, and IRNI(CR) and verify their increase in expressivity. We do so by applying them to synthetically crafted, hard data sets as well as standard practical data sets. Furthermore, we propose an automated training approach used throughout the benchmarks.

**Network architectures and optimization.** For all experiments, we use the same general architecture, the Adam optimizer, and use the area under the receiver operating characteristic (AUROC). We optimize each method using a bayesian hyperparameter search in the same hyperparameter space. To estimate the performance, we use Monte Carlo cross-validation in an outer test loop and an inner validation loop estimating nested $10 \times 9$-fold cross-validation. The bayesian hyperparameter search is capped at evaluating 50 points in hyperparameter space. To encourage the models to optimize faster as well as to avoid overfitting, we add a penalty to the AUROC estimate based on some hyperparameters. The reported test AUROC does not include these penalties. To compute the node sequence for IRNI(CR) as well as the color refinement for CLIP we use DEJAVU (Anders & Schweitzer, 2021a). These choices are specified further in the appendix. In the following, we refer to a GIN without any URF as None, while we refer to a GIN with some initialization as RNI, CLIP, RP, or IRNI(CR) depending on the URF that is used. Each of these methods is also limited in the number of dimensions added.

**Data sets.** We evaluate different models on datasets used in prior work on URF, specifically EXP, CEXP, TRI, TRIX, CSL, PROTEINS, MUTAG, and NCI1 (Srinivasan et al.; Borgwardt et al., 2005; Wale & Karypis, 2006; Murphy et al., 2019; Sato et al., 2021; Abboud et al., 2021). EXP, CEXP, TRI, TRIX, and CSL are synthetic data sets made up of graphs not distinguishable by the color refinement algorithm. TRI and TRIX contain 3-regular graphs and use the same training set while differing in the test set. The task is to detect triangles. EXP and CEXP consist of graphs carefully constructed so that each graph is in a pair that is indistinguishable by color refinement while encoding a satisfiable and unsatisfiable formula respectively. For CEXP 50% of all satisfiable graphs are modified to be distinguishable by color refinement from their unsatisfiable counterparts. CSL consists of 41-cycles with regular skip-connections according to 10 co-primes of 41. Each co-prime defines one class for the CSL task. The results of this experiment can be found in Table 1.

# 6 Discussion

We notice that on the synthetic hard datasets TRI, TRIX, EXP, CEXP, and CSL, all methods improve the discriminatory power compared to not using any form of URF. We now compare the methods based on the three primary parameters encoding, ensembling, and refinement.

Concerning the encoding on TRIX and CSL, there seem to be noticeable differences between RNI and the other methods. The poorer performance of CLIP, RP, and IRNI(CR) on TRIX is easily explained. Since the task is to detect triangles locally in the graph and the graph is regular, an individualization is required close to the triangle to be able to detect it. RNI individualizes everywhere simultaneously, while CLIP, RP, and IRNI(CR) only individualize locally. This means the detection of triangles depends on random chance. EoR helps since it increases this chance. The difference on CSL is not so clearly explainable. We do not know why RNI performs significantly worse here. Looking more specifically at the difference between RP and RNI, we remark that RP substantially outperforms RNI on PROTEINS, while RNI outperforms RP on MUTAG.

EoR appears almost always to improve the performance. As such, we would advise its consideration whenever one of these URF is used in practice. Notice that its use does not increase training time and only linearly increases prediction time. EoR also appears to guarantee that at least one of RNI, CLIP, RP, or IRNI(CR) will outperform models without this expressibility increase.

Considering the methods that use additional refinement to reduce the introduced randomness, namely CLIP and IRNI(CR), we observe they outperform RP on MUTAG, while RP outperforms the other two on PROTEINS and NCI1. On the synthetic hard datasets, in particular with ensembling, the three methods perform very similarly.

Overall, no method appears to be the uniformly best method for practical use. We suspect that overfitting to the different features introduced by RNI, CLIP, RP, and IRNI(CR) plays a significant role in which of these URF performs best.

## 7    Conclusion

We introduced IR as it applies to machine learning in the form of the IRNI framework. This enables the development of many URF based on selecting refinement, cell selector, and how to encode individualizations into the network. No URF introduced so far is the clear front runner. However, the BHO tuning approach presented here feasibly allows for the optimization of the IRNI hyperparameters in addition to other model hyperparameters. The IRNI hyperparameters also serve the unifying IRNI umbrella, as we were able to describe all existing and new URF based on these. Moreover, IRNI has a rigorous theoretical foundation ensuring equivariance and universality. We hope this will aid in systematic improvements in future research regarding GNNs. Practically, our findings imply that for each new graph learning task all mentioned URF need to be evaluated to determine the best fit.

## Acknowledgement

We thank the reviewers for their constructive feedback that helped improve the paper. The authors acknowledge support by the Carl-Zeiss Foundation, the BMWK award 01MK20014U, the DFG awards KL 2698/2-1, KL 2698/5-1, KL 2698/6-1 and KL 2698/7-1,and the BMBF awards 01|S18051A, 03|B0770E, and 01|S21010C. The research leading to these results has received funding from the European Research Council (ERC) under the European Union's Horizon 2020 research and innovation programme (EngageS: grant agreement No. 820148).

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

## A  Refinement Definition

We discuss the slight technicality in our definition of refinement in the individualization-refinement paradigm. Compared to McKay & Piperno (2014), we are lacking the requirement that $\pi' = \mathrm{Ref}(G, \pi, \nu)$ must be finer than $\pi$, making our definition of refinements slightly more general. With respect to the results, this implies that for (non-trivially) colored graphs, discrete refined colorings might not respect the initial coloring. This can potentially lead to issues with automorphisms and canonization since, essentially, the isomorphism invariant implied by the refinement is too weak. However, there is a simple fix using other components of the IR framework: we make use of invariants, which are not immediately relevant for IRNI itself. Whenever a coloring $\pi'$ of a node $\nu$ in the tree is discrete, we can use the complete isomorphism invariant $(G^{\pi'}, \pi^{\pi'})$ (note that since $\pi'$ is discrete, it also defines a permutation on the vertices of $G$). I.e., we identity the node $\nu$ with the invariant $(G^{\pi'}, \pi^{\pi'})$. As mentioned above, this does not influence any of the results of this paper directly – it only serves to make the description sound in terms of the further discussion in McKay & Piperno (2014). We refer to McKay & Piperno (2014) for an in-depth discussion of invariants.

## B  Relational Pooling

Relational pooling (RP) Murphy et al. (2019) is more general than discussed in this paper. In its initial formulation $\frac{1}{|V|!} \sum_{\pi \in S_{|V|}} f(G^\pi, X^\pi)$ it is not tractable and does thus not fit into the comparison considered

Table 2: The hyperparameter space for bayesian hyperparameter optimization. Penalty describes how the parameter influences the hyperparameter optimization.

| Parameter | batch size | epochs | learning rate | weight decay | features | layers | dimensions | step size | EoR |
|---|---|---|---|---|---|---|---|---|---|
| Minimum | 8 | 32 | 1e-6 | 1e-10 | 16 | 2 | 1 | 0.01 | 1 |
| Maximum | 256 | 512 | 1e-2 | 1e-0.3 | 128 | 10 | 5 | 1.0 | 64 |
| Data-Type | Integer | Integer | Real | Real | Integer | Integer | Integer | Real | Integer |
| Penalty | 1-(x/256) | x/512 | None | None | x/128 | x/10 | None | None | x/64 |

here. In the original paper, three methods to make RP tractable are discussed. The first uses canonization, which itself is generally not tractable, and thus is not considered. The third uses a fixed point tractable (FPT) formulation (essentially describing a less expressive version of the $k$-dimensional WL algorithm), which is generally not universal unless the FPT parameter is not fixed, and even then its costs are polynomial in the FPT parameter which is computationally too expensive for this comparison. Only the second discussed option, which is originally motivated by stochastically estimating the initial formulation, is both universal and computationally practical, which is why it is considered here. In particular, it is also the only version implemented and tested in the original paper.

## C   Edge Coloring

We discuss the differences when using edge colors since the MUTAG data set contains edge labels, which we consider in the experiments.

In order to handle edge colors in DEJAVU for color refinement and random IR walks, we made the following modifications. We encode edge colors as vertex colors: we subdivide each edge of the graph using a vertex and color that vertex according to the color of the edge. Then, we ensured that the cell selector never selects nodes that were inserted to subdivide an edge, i.e., the solver can still only individualize vertices that truly correspond to the original vertices of the graph.

We want to note that there is a more efficient albeit more involved way of resolving the issue Piperno (2018).

## D   Color Encoding

We discuss precisely how we encoded colors for the data sets discussed in this work. First, notice that we need to encode labels for nodes and for edges. We used the same method to encode colors for both. Almost all node and edge features can be considered binary numbers due to how the data sets are encoded. The only exception to this rule is the PROTEINS data set which has one natural number followed by a binary number. Thus, we simply read the node and edge features as a binary number, where the first bit has the highest order, and the last bit has the lowest order. This ensures that node and edge features are encoded as different natural numbers if the original features were different. However, due to the size of natural numbers, that specifically the NCI1 dataset produces, as it has 37 node features, we also apply a modulo operation by 12345.

## E   Experiments

**Network architecture.**   Each MLP in each GIN layer has three layers (input, hidden, output) that widen to a fixed number of features as soon as possible and remain there throughout all subsequent layers. The input and hidden layers of every MLP are followed by a batch-norm operation. For graph classification tasks, we use global mean pooling followed by a linear transform and dropout with $p = 0.5$. We use the node embeddings after each GIN layer in this way and sum over all of them for the final graph representation. As activation functions, we use only ReLU. For node-classification tasks, we do the same as before without the global mean pooling.

**Bayesian hyperparameter optimization.** We optimize over the batch size, the number of epochs, the learning rate, weight decay, the number of features the GIN layers expand to and operate on, the number of GIN layers, the number of dimensions added for the node initializations, the fraction of epochs after which the learning rate gets decreased by 0.5 consecutively, and the number of random samples for EoR. Table 2 describes the hyperparameter space used for the experiments. For in indepth description of each parameter:

- Batch size refers to the batch size used during training. The smaller it is, the more it penalizes the evaluation metric during the bayesian hyperparameter search.

- Epochs refers to the number of epochs used during training. The bigger they are, the more they penalize.

- Learning rate refers to the initial learning rate used inside of the Adam optimizer.

- Weight decay refers to the typical weight decay parameter inside of the Adam optimizer.

- Features refer to the number of features the MLPs inside of the GIN layers expand to. Each node will have features many features after the first GIN layers first MLP layer. The more features, the more they penalize.

- Layers refer to the number of layers of the GIN networks. The more layers, the more they penalize.

- Dimensions refer to the number of dimensions each node's features are expanded by for the new features introduced by the different URF methods. This parameter is the same as the d in d-IRNI.

- Step size is a relative parameter that influences how the learning rate is changed during training depending on the number of epochs. For instance, if the number of epochs is set at 100, then a step size of 0.1 will mean the learning rate is divided by 0.5 every 10 epochs, a step size of 0.5 would mean the learning rate is divided by 0.5 once after 50 epochs, and a step size of 1 indicates the learning rate is never dropped.

- EoR refers to the number of random samples that are used to estimate the output of the network. For instance, for an EoR of 10 the input is modified 10 times i.i.d using the URF of choice. All 10 inputs are then passed through the network and the final predictions are then averaged for all 10 outputs.

Tables 4, ..., 19 show the mean and standard deviation of the best-found hyperparameters across their seeds for all the datasets and methods. For the learning rate and weight decay, only the exponent is given.

**Monte Carlo cross-validation.** Each dataset is split using stratified 10-fold cross-validation with random shuffling. The first fold is used as the test set and the 9 others are used for bayesian hyperparameter optimization. After the best model is found, it is trained on all 9 folds and its performance on the test set is reported. This is repeated 10 times with different random shuffles. If the dataset initially already provides a test set, then the bayesian hyperparameter optimization is repeated for 10 different seeds. In the inner loop, which we referred to before as just bayesian hyperparameter optimization, the data is split using stratified 9-fold cross-validation. The first fold is used as a validation set and the other 8 folds are used to train the model, after which its performance is reported on the validation set. This is repeated 3 times with different random shuffles. This essentially estimates nested $10 \times 9$-fold cross-validation. The 10 and 3 were chosen based on a time budget. The performance on PROTEINS, MUTAG, and NCI1 is particularly sensitive to the variance in the performance estimate, so we expect to see an improved performance if more estimates are used.

**Test system and time budget.** The system that was used to do the experiments mentioned in this work is made up of:

- #60-Ubuntu SMP Tue Jul 2 18:22:20 UTC 2019 4.15.0-55-generic

Table 3: The time in seconds for the entire evaluation process of 1 seed of a GIN network with RNI, CLIP, RP, IRNI(CR), and without any of these on selected data sets. $_{EoR}$ indicates the use of ensembling over randomness.

| Method | PROTEINS | MUTAG | NCI1 | TRI | TRIX | EXP | CEXP | CSL |
|---|---|---|---|---|---|---|---|---|
| None | $25672 \pm 5426$ | $7275 \pm 5532$ | $121487 \pm 63237$ | $10878 \pm 2328$ | $10809 \pm 2288$ | $20735 \pm 9345$ | $19652 \pm 5739$ | $2286 \pm 409$ |
| RNI | $34499 \pm 16968$ | $7802 \pm 2471$ | $101154 \pm 29446$ | $53897 \pm 27607$ | $46855 \pm 21908$ | $53381 \pm 35114$ | $58668 \pm 19114$ | $9999 \pm 4512$ |
| CLIP | $36189 \pm 10374$ | $12268 \pm 3955$ | $151603 \pm 36904$ | $52437 \pm 19898$ | $48608 \pm 20161$ | $65597 \pm 13000$ | $55558 \pm 13902$ | $10697 \pm 3146$ |
| RP | $26042 \pm 8782$ | $8783 \pm 2234$ | $126282 \pm 36043$ | $56640 \pm 35035$ | $48930 \pm 34016$ | $52605 \pm 29282$ | $53750 \pm 25979$ | $6321 \pm 1917$ |
| IRNI | $33383 \pm 13685$ | $7793 \pm 1560$ | $156347 \pm 35831$ | $42454 \pm 13367$ | $40597 \pm 15643$ | $64123 \pm 13412$ | $50042 \pm 14155$ | $10937 \pm 1895$ |
| $RNI_{EoR}$ | $26593 \pm 9390$ | $7810 \pm 3740$ | $106401 \pm 30762$ | $26767 \pm 10270$ | $25630 \pm 14628$ | $34604 \pm 12037$ | $33747 \pm 9654$ | $8832 \pm 6506$ |
| $CLIP_{EoR}$ | $65225 \pm 31132$ | $8418 \pm 2048$ | $118096 \pm 17091$ | $24487 \pm 4718$ | $28594 \pm 6917$ | $38565 \pm 7954$ | $82525 \pm 17467$ | $6810 \pm 1807$ |
| $RP_{EoR}$ | $22203 \pm 4547$ | $6309 \pm 2749$ | $117571 \pm 43888$ | $21233 \pm 5526$ | $22385 \pm 7073$ | $31831 \pm 13730$ | $37978 \pm 12934$ | $6513 \pm 2271$ |
| $IRNI_{EoR}$ | $33337 \pm 12360$ | $8204 \pm 3483$ | $160859 \pm 55380$ | $33799 \pm 4711$ | $61533 \pm 21959$ | $52990 \pm 13094$ | $75701 \pm 18933$ | $5528 \pm 1360$ |

Table 4: The mean and standard deviation for each found hyperparameter for each method on the PRO-TEINS dataset without EoR

| Parameter | batch size | epochs | learning rate | weight decay | features | layers | dimensions | step size |
|---|---|---|---|---|---|---|---|---|
| None | $152.70 \pm 83.17$ | $278.10 \pm 185.29$ | $-3.21 \pm 0.76$ | $-5.12 \pm 2.72$ | $76.00 \pm 35.85$ | $5.80 \pm 1.78$ | $3.30 \pm 1.68$ | $0.53 \pm 0.32$ |
| RNI | $86.30 \pm 82.32$ | $303.50 \pm 131.03$ | $-3.53 \pm 0.89$ | $-5.72 \pm 2.14$ | $59.50 \pm 41.13$ | $5.30 \pm 3.13$ | $3.30 \pm 1.27$ | $0.63 \pm 0.29$ |
| CLIP | $141.90 \pm 93.62$ | $224.20 \pm 140.10$ | $-3.61 \pm 1.06$ | $-7.41 \pm 2.70$ | $60.40 \pm 38.66$ | $3.80 \pm 1.33$ | $3.60 \pm 1.50$ | $0.58 \pm 0.30$ |
| RP | $192.70 \pm 52.98$ | $262.50 \pm 162.44$ | $-2.90 \pm 0.66$ | $-6.76 \pm 3.28$ | $55.00 \pm 35.34$ | $2.80 \pm 1.17$ | $4.10 \pm 1.04$ | $0.55 \pm 0.33$ |
| IRNI | $141.70 \pm 90.51$ | $308.30 \pm 165.17$ | $-2.97 \pm 1.17$ | $-7.45 \pm 2.51$ | $72.40 \pm 48.34$ | $3.00 \pm 1.34$ | $3.20 \pm 1.47$ | $0.45 \pm 0.25$ |

- 2 Intel(R) Xeon(R) Gold 6154 CPU @ 3.00GHz

- 754GiB System memory

- 10 GeForce RTX 2080 Ti

The experiments took approximately 20 days without testing and approximately 1 month with testing, where the machine was not used on full load always. Table 3 shows the computation time in seconds for 1 seed for each of the methods on each separate dataset.

**Python libraries.** We used python 3.8.10 to implement all the models and conduct all the experiments:

- dejavu-gi 0.1.3 (for IRNI(CR))

- networkx 2.6.3 (for constructing TRI and TRIX)

- numpy 1.21.4

- scikit-learn 1.0.1

- scikit-optimize 0.9.0 (for Bayesian hyperparameter optimization)

- scipy 1.7.3

- torch 1.10.0

- torch-geometric 2.0.2 (specifically for graph related machine learning)

**Random Seed** For the experiments we used the random seeds 0 through 9 as input to our code. However, our experiments might not be perfectly reproducible as DEJAVU the package we use to calculate random IR paths does not allow for its seed to be set.

Table 5: The mean and standard deviation for each found hyperparameter for each method on the MUTAG dataset without EoR

| Parameter | batch size | epochs | learning rate | weight decay | features | layers | dimensions | step size |
|---|---|---|---|---|---|---|---|---|
| None | $157.10 \pm 78.41$ | $329.80 \pm 152.31$ | $-3.67 \pm 0.63$ | $-6.36 \pm 2.25$ | $72.30 \pm 32.85$ | $5.70 \pm 3.07$ | $2.90 \pm 1.51$ | $0.61 \pm 0.30$ |
| RNI | $145.10 \pm 90.67$ | $315.40 \pm 150.22$ | $-3.25 \pm 0.78$ | $-8.43 \pm 2.22$ | $55.40 \pm 30.11$ | $5.80 \pm 2.79$ | $3.60 \pm 1.43$ | $0.76 \pm 0.24$ |
| CLIP | $142.20 \pm 94.73$ | $208.90 \pm 139.70$ | $-3.11 \pm 0.91$ | $-7.12 \pm 2.84$ | $102.30 \pm 17.73$ | $6.30 \pm 2.93$ | $2.90 \pm 1.14$ | $0.68 \pm 0.27$ |
| RP | $181.90 \pm 64.39$ | $256.20 \pm 131.57$ | $-3.07 \pm 0.77$ | $-6.27 \pm 3.09$ | $55.10 \pm 29.18$ | $6.60 \pm 2.97$ | $3.80 \pm 0.98$ | $0.61 \pm 0.30$ |
| IRNI | $167.20 \pm 78.88$ | $371.80 \pm 120.38$ | $-3.21 \pm 1.09$ | $-5.81 \pm 3.25$ | $68.10 \pm 40.89$ | $5.80 \pm 3.09$ | $2.80 \pm 1.47$ | $0.62 \pm 0.27$ |

Table 6: The mean and standard deviation for each found hyperparameter for each method on the NCI1 dataset without EoR

| Parameter | batch size | epochs | learning rate | weight decay | features | layers | dimensions | step size | |
|---|---|---|---|---|---|---|---|---|---|
| None | $75.20 \pm 60.04$ | $270.60 \pm 138.40$ | $-3.78 \pm 0.85$ | $-5.43 \pm 2.02$ | $91.10 \pm 28.83$ | $7.00 \pm 2.05$ | $3.40 \pm 1.11$ | $0.65 \pm 0.32$ | $1.00 \pm 0.00$ |
| RNI | $103.70 \pm 59.22$ | $309.80 \pm 84.42$ | $-2.97 \pm 0.73$ | $-5.22 \pm 1.99$ | $89.50 \pm 32.23$ | $6.00 \pm 1.55$ | $3.00 \pm 1.73$ | $0.62 \pm 0.23$ | $1.00 \pm 0.00$ |
| CLIP | $131.75 \pm 58.16$ | $403.25 \pm 117.57$ | $-3.28 \pm 0.79$ | $-5.79 \pm 1.93$ | $97.38 \pm 30.59$ | $7.38 \pm 1.58$ | $2.50 \pm 1.66$ | $0.50 \pm 0.19$ | $1.00 \pm 0.00$ |
| ORNI | $157.00 \pm 81.81$ | $362.70 \pm 134.81$ | $-3.15 \pm 0.72$ | $-6.06 \pm 2.20$ | $69.00 \pm 22.95$ | $5.40 \pm 1.69$ | $3.30 \pm 1.19$ | $0.45 \pm 0.21$ | $1.00 \pm 0.00$ |
| IRNI | $93.12 \pm 83.31$ | $369.38 \pm 93.77$ | $-3.88 \pm 0.61$ | $-6.40 \pm 2.27$ | $105.50 \pm 26.80$ | $8.12 \pm 2.32$ | $1.12 \pm 0.33$ | $0.57 \pm 0.29$ | $1.00 \pm 0.00$ |

Table 7: The mean and standard deviation for each found hyperparameter for each method on the TRI dataset without EoR

| Parameter | batch size | epochs | learning rate | weight decay | features | layers | dimensions | step size |
|---|---|---|---|---|---|---|---|---|
| None | $256.00 \pm 0.00$ | $32.00 \pm 0.00$ | $-4.15 \pm 1.90$ | $-4.15 \pm 4.71$ | $16.00 \pm 0.00$ | $2.00 \pm 0.00$ | $3.80 \pm 1.83$ | $0.50 \pm 0.49$ |
| RNI | $132.50 \pm 91.25$ | $454.30 \pm 55.92$ | $-2.61 \pm 0.48$ | $-8.01 \pm 2.22$ | $92.10 \pm 26.43$ | $9.50 \pm 0.67$ | $1.60 \pm 1.28$ | $0.66 \pm 0.29$ |
| CLIP | $129.30 \pm 105.04$ | $436.60 \pm 52.67$ | $-2.68 \pm 0.31$ | $-7.31 \pm 2.06$ | $78.60 \pm 25.15$ | $8.40 \pm 1.36$ | $3.80 \pm 0.75$ | $0.54 \pm 0.33$ |
| RP | $161.40 \pm 102.91$ | $397.30 \pm 98.27$ | $-2.55 \pm 0.38$ | $-7.80 \pm 1.65$ | $80.60 \pm 21.90$ | $8.80 \pm 1.40$ | $4.40 \pm 0.80$ | $0.56 \pm 0.22$ |
| IRNI | $147.70 \pm 93.12$ | $437.30 \pm 87.21$ | $-2.67 \pm 0.41$ | $-6.83 \pm 2.14$ | $78.30 \pm 27.22$ | $8.10 \pm 1.51$ | $3.80 \pm 0.98$ | $0.68 \pm 0.29$ |

Table 8: The mean and standard deviation for each found hyperparameter for each method on the TRIX dataset without EoR

| Parameter | batch size | epochs | learning rate | weight decay | features | layers | dimensions | step size |
|---|---|---|---|---|---|---|---|---|
| None | $256.00 \pm 0.00$ | $32.00 \pm 0.00$ | $-4.15 \pm 1.90$ | $-4.15 \pm 4.71$ | $16.00 \pm 0.00$ | $2.00 \pm 0.00$ | $3.80 \pm 1.83$ | $0.50 \pm 0.49$ |
| RNI | $126.40 \pm 67.20$ | $431.20 \pm 43.66$ | $-2.62 \pm 0.30$ | $-7.63 \pm 2.11$ | $94.80 \pm 24.86$ | $9.40 \pm 0.80$ | $2.00 \pm 1.34$ | $0.46 \pm 0.19$ |
| CLIP | $120.70 \pm 83.98$ | $378.60 \pm 77.44$ | $-2.65 \pm 0.45$ | $-7.15 \pm 1.79$ | $67.90 \pm 16.12$ | $8.10 \pm 1.45$ | $4.00 \pm 0.89$ | $0.64 \pm 0.26$ |
| RP | $168.20 \pm 96.77$ | $445.90 \pm 54.74$ | $-2.63 \pm 0.44$ | $-6.80 \pm 1.63$ | $80.40 \pm 32.04$ | $8.50 \pm 1.12$ | $4.40 \pm 1.02$ | $0.69 \pm 0.28$ |
| IRNI | $185.10 \pm 83.23$ | $451.80 \pm 75.61$ | $-2.59 \pm 0.31$ | $-9.16 \pm 1.23$ | $76.40 \pm 29.88$ | $8.90 \pm 1.30$ | $4.10 \pm 0.94$ | $0.63 \pm 0.21$ |

Table 9: The mean and standard deviation for each found hyperparameter for each method on the EXP dataset without EoR

| Parameter | batch size | epochs | learning rate | weight decay | features | layers | dimensions | step size |
|---|---|---|---|---|---|---|---|---|
| None | $205.90 \pm 67.09$ | $172.10 \pm 166.92$ | $-4.29 \pm 1.82$ | $-4.40 \pm 3.54$ | $52.10 \pm 40.55$ | $4.30 \pm 2.33$ | $2.60 \pm 1.80$ | $0.56 \pm 0.39$ |
| RNI | $164.40 \pm 84.80$ | $425.10 \pm 79.47$ | $-2.95 \pm 0.50$ | $-7.35 \pm 1.75$ | $63.00 \pm 37.26$ | $8.80 \pm 1.40$ | $2.30 \pm 1.68$ | $0.64 \pm 0.28$ |
| CLIP | $223.20 \pm 51.71$ | $307.30 \pm 120.73$ | $-2.78 \pm 0.59$ | $-7.11 \pm 2.50$ | $63.50 \pm 41.48$ | $7.60 \pm 1.20$ | $1.80 \pm 1.17$ | $0.42 \pm 0.23$ |
| RP | $139.10 \pm 86.74$ | $378.10 \pm 115.58$ | $-3.59 \pm 0.40$ | $-7.03 \pm 2.21$ | $84.90 \pm 41.42$ | $8.80 \pm 1.25$ | $4.70 \pm 0.46$ | $0.52 \pm 0.24$ |
| IRNI | $174.50 \pm 78.69$ | $210.80 \pm 159.69$ | $-3.20 \pm 0.38$ | $-7.32 \pm 2.63$ | $33.20 \pm 12.40$ | $8.20 \pm 1.54$ | $1.80 \pm 1.08$ | $0.61 \pm 0.25$ |

Table 10: The mean and standard deviation for each found hyperparameter for each method on the CEXP dataset without EoR

| Parameter | batch size | epochs | learning rate | weight decay | features | layers | dimensions | step size |
|---|---|---|---|---|---|---|---|---|
| None | $149.70 \pm 86.61$ | $143.60 \pm 98.14$ | $-2.51 \pm 0.54$ | $-5.26 \pm 2.77$ | $49.20 \pm 30.00$ | $6.80 \pm 2.23$ | $2.70 \pm 1.42$ | $0.33 \pm 0.25$ |
| RNI | $153.40 \pm 86.97$ | $430.10 \pm 102.38$ | $-3.69 \pm 0.31$ | $-6.46 \pm 2.01$ | $63.40 \pm 36.21$ | $9.10 \pm 1.45$ | $1.40 \pm 0.92$ | $0.87 \pm 0.14$ |
| CLIP | $190.50 \pm 49.88$ | $308.00 \pm 90.24$ | $-3.38 \pm 0.77$ | $-7.61 \pm 2.04$ | $54.30 \pm 30.84$ | $8.30 \pm 1.55$ | $2.10 \pm 1.37$ | $0.66 \pm 0.23$ |
| RP | $130.30 \pm 85.78$ | $353.30 \pm 99.67$ | $-3.65 \pm 0.28$ | $-5.25 \pm 2.11$ | $75.90 \pm 27.93$ | $8.80 \pm 1.08$ | $4.60 \pm 0.66$ | $0.53 \pm 0.27$ |
| IRNI | $169.90 \pm 87.46$ | $245.80 \pm 157.52$ | $-3.13 \pm 0.89$ | $-6.69 \pm 1.85$ | $49.00 \pm 34.81$ | $8.00 \pm 2.05$ | $2.80 \pm 1.54$ | $0.51 \pm 0.30$ |

Table 11: The mean and standard deviation for each found hyperparameter for each method on the CSL dataset without EoR

| Parameter | batch size | epochs | learning rate | weight decay | features | layers | dimensions | step size |
|---|---|---|---|---|---|---|---|---|
| None | $256.00 \pm 0.00$ | $32.00 \pm 0.00$ | $-4.15 \pm 1.90$ | $-4.15 \pm 4.71$ | $16.00 \pm 0.00$ | $2.00 \pm 0.00$ | $3.80 \pm 1.83$ | $0.50 \pm 0.49$ |
| RNI | $74.90 \pm 86.57$ | $481.70 \pm 58.11$ | $-3.15 \pm 0.45$ | $-5.27 \pm 2.27$ | $90.00 \pm 35.89$ | $8.30 \pm 1.35$ | $1.70 \pm 1.42$ | $0.62 \pm 0.26$ |
| CLIP | $179.10 \pm 73.78$ | $312.10 \pm 91.59$ | $-2.25 \pm 0.35$ | $-6.33 \pm 2.00$ | $66.40 \pm 36.34$ | $7.00 \pm 1.67$ | $3.50 \pm 1.12$ | $0.58 \pm 0.18$ |
| RP | $185.00 \pm 70.37$ | $319.30 \pm 88.94$ | $-2.50 \pm 0.48$ | $-6.31 \pm 2.48$ | $53.00 \pm 29.85$ | $7.50 \pm 1.43$ | $4.20 \pm 0.87$ | $0.42 \pm 0.24$ |
| IRNI | $191.90 \pm 58.75$ | $240.00 \pm 114.25$ | $-2.86 \pm 0.53$ | $-6.67 \pm 2.83$ | $37.70 \pm 14.21$ | $4.70 \pm 1.62$ | $4.00 \pm 0.89$ | $0.41 \pm 0.23$ |

Table 12: The mean and standard deviation for each found hyperparameter for each method on the PROTEINS dataset with EoR

| Parameter | batch size | epochs | learning rate | weight decay | features | layers | dimensions | step size | EoR |
|---|---|---|---|---|---|---|---|---|---|
| RNI | $180.80 \pm 72.66$ | $265.90 \pm 147.07$ | $-3.98 \pm 1.03$ | $-6.88 \pm 2.31$ | $62.10 \pm 33.05$ | $5.60 \pm 2.87$ | $4.60 \pm 0.49$ | $0.67 \pm 0.30$ | $36.10 \pm 17.31$ |
| CLIP | $92.10 \pm 69.64$ | $194.30 \pm 153.41$ | $-3.42 \pm 1.00$ | $-3.85 \pm 2.41$ | $63.70 \pm 37.81$ | $5.90 \pm 2.77$ | $3.60 \pm 1.20$ | $0.45 \pm 0.30$ | $21.00 \pm 17.40$ |
| RP | $157.00 \pm 78.16$ | $281.10 \pm 153.80$ | $-2.50 \pm 0.28$ | $-4.69 \pm 3.19$ | $66.00 \pm 39.44$ | $2.80 \pm 1.17$ | $2.90 \pm 1.04$ | $0.41 \pm 0.25$ | $41.60 \pm 17.45$ |
| IRNI | $163.70 \pm 74.50$ | $296.30 \pm 169.70$ | $-2.83 \pm 1.00$ | $-7.44 \pm 2.49$ | $57.50 \pm 46.20$ | $3.90 \pm 1.76$ | $3.80 \pm 1.25$ | $0.43 \pm 0.41$ | $18.50 \pm 13.34$ |

Table 13: The mean and standard deviation for each found hyperparameter for each method on the MUTAG dataset with EoR

| Parameter | batch size | epochs | learning rate | weight decay | features | layers | dimensions | step size | EoR |
|---|---|---|---|---|---|---|---|---|---|
| RNI | $110.70 \pm 80.39$ | $221.50 \pm 132.31$ | $-3.57 \pm 0.69$ | $-6.83 \pm 2.79$ | $78.20 \pm 35.47$ | $6.40 \pm 2.69$ | $2.90 \pm 1.45$ | $0.59 \pm 0.24$ | $34.40 \pm 15.88$ |
| CLIP | $111.70 \pm 88.57$ | $269.30 \pm 114.02$ | $-3.65 \pm 0.61$ | $-4.71 \pm 3.12$ | $80.30 \pm 34.81$ | $6.90 \pm 2.21$ | $2.80 \pm 1.25$ | $0.42 \pm 0.36$ | $14.50 \pm 14.12$ |
| RP | $153.10 \pm 79.12$ | $255.70 \pm 173.48$ | $-2.96 \pm 0.89$ | $-5.14 \pm 2.90$ | $84.20 \pm 46.18$ | $7.30 \pm 2.49$ | $2.70 \pm 1.10$ | $0.68 \pm 0.29$ | $35.80 \pm 16.77$ |
| IRNI | $158.40 \pm 84.20$ | $310.00 \pm 129.19$ | $-3.65 \pm 1.01$ | $-4.81 \pm 2.33$ | $76.00 \pm 43.38$ | $5.60 \pm 3.14$ | $2.10 \pm 1.37$ | $0.61 \pm 0.18$ | $39.30 \pm 21.87$ |

Table 14: The mean and standard deviation for each found hyperparameter for each method on the NCI1 dataset with EoR

| Parameter | batch size | epochs | learning rate | weight decay | features | layers | dimensions | step size | EoR |
|---|---|---|---|---|---|---|---|---|---|
| RNI | $153.00 \pm 86.19$ | $266.60 \pm 120.89$ | $-4.46 \pm 0.33$ | $-5.09 \pm 1.85$ | $96.10 \pm 22.78$ | $8.20 \pm 1.54$ | $3.20 \pm 1.17$ | $0.56 \pm 0.29$ | $44.60 \pm 17.35$ |
| CLIP | $162.50 \pm 76.87$ | $383.90 \pm 109.63$ | $-4.04 \pm 0.77$ | $-7.20 \pm 2.36$ | $86.70 \pm 35.18$ | $7.90 \pm 1.87$ | $2.40 \pm 1.20$ | $0.41 \pm 0.31$ | $35.30 \pm 14.49$ |
| RP | $138.10 \pm 85.82$ | $246.50 \pm 59.54$ | $-4.54 \pm 0.41$ | $-5.02 \pm 2.15$ | $102.60 \pm 26.59$ | $7.00 \pm 1.90$ | $3.60 \pm 0.80$ | $0.55 \pm 0.30$ | $43.00 \pm 10.61$ |
| IRNI | $174.10 \pm 104.05$ | $356.70 \pm 137.42$ | $-3.68 \pm 0.47$ | $-7.05 \pm 2.42$ | $102.60 \pm 18.67$ | $7.50 \pm 2.01$ | $1.80 \pm 0.98$ | $0.61 \pm 0.33$ | $7.50 \pm 12.67$ |

Table 15: The mean and standard deviation for each found hyperparameter for each method on the TRI dataset with EoR

| Parameter | batch size | epochs | learning rate | weight decay | features | layers | dimensions | step size | EoR |
|---|---|---|---|---|---|---|---|---|---|
| RNI | $200.40 \pm 56.35$ | $236.70 \pm 85.80$ | $-2.54 \pm 0.52$ | $-5.66 \pm 2.51$ | $51.90 \pm 29.79$ | $5.90 \pm 1.87$ | $3.30 \pm 1.19$ | $0.58 \pm 0.22$ | $37.90 \pm 17.17$ |
| CLIP | $205.40 \pm 43.87$ | $219.50 \pm 118.76$ | $-2.79 \pm 0.48$ | $-5.77 \pm 2.64$ | $38.90 \pm 25.93$ | $4.50 \pm 1.63$ | $4.00 \pm 1.18$ | $0.58 \pm 0.28$ | $25.90 \pm 16.35$ |
| RP | $214.00 \pm 47.44$ | $214.90 \pm 137.52$ | $-2.43 \pm 0.45$ | $-7.03 \pm 2.55$ | $31.30 \pm 22.45$ | $5.30 \pm 2.28$ | $4.40 \pm 0.66$ | $0.44 \pm 0.27$ | $25.30 \pm 15.07$ |
| IRNI | $198.30 \pm 69.54$ | $170.00 \pm 139.94$ | $-2.72 \pm 0.73$ | $-6.79 \pm 2.77$ | $30.70 \pm 12.45$ | $3.90 \pm 1.45$ | $3.60 \pm 1.62$ | $0.31 \pm 0.29$ | $34.40 \pm 15.81$ |

Table 16: The mean and standard deviation for each found hyperparameter for each method on the TRIX dataset with EoR

| Parameter | batch size | epochs | learning rate | weight decay | features | layers | dimensions | step size | EoR |
|---|---|---|---|---|---|---|---|---|---|
| RNI | $203.30 \pm 60.54$ | $264.40 \pm 93.60$ | $-2.50 \pm 0.44$ | $-6.91 \pm 2.52$ | $45.20 \pm 22.30$ | $5.70 \pm 2.33$ | $3.20 \pm 1.08$ | $0.54 \pm 0.24$ | $31.60 \pm 14.83$ |
| CLIP | $187.10 \pm 73.91$ | $140.70 \pm 97.46$ | $-2.94 \pm 0.54$ | $-6.43 \pm 3.27$ | $49.60 \pm 29.86$ | $4.30 \pm 1.35$ | $3.80 \pm 1.33$ | $0.62 \pm 0.32$ | $36.50 \pm 16.41$ |
| RP | $190.20 \pm 78.03$ | $182.80 \pm 124.46$ | $-3.16 \pm 0.68$ | $-5.02 \pm 2.93$ | $53.10 \pm 27.94$ | $3.80 \pm 1.54$ | $4.00 \pm 1.18$ | $0.64 \pm 0.18$ | $32.20 \pm 18.42$ |
| IRNI | $199.20 \pm 57.18$ | $187.90 \pm 111.60$ | $-2.60 \pm 0.62$ | $-5.01 \pm 2.37$ | $52.30 \pm 21.19$ | $5.50 \pm 2.25$ | $4.20 \pm 1.17$ | $0.55 \pm 0.27$ | $26.10 \pm 16.02$ |

Table 17: The mean and standard deviation for each found hyperparameter for each method on the EXP dataset with EoR

| Parameter | batch size | epochs | learning rate | weight decay | features | layers | dimensions | step size | EoR |
|---|---|---|---|---|---|---|---|---|---|
| RNI | $221.50 \pm 53.90$ | $359.50 \pm 91.93$ | $-3.35 \pm 0.58$ | $-6.44 \pm 3.08$ | $63.60 \pm 35.13$ | $7.70 \pm 1.35$ | $2.50 \pm 1.28$ | $0.68 \pm 0.35$ | $29.30 \pm 20.25$ |
| CLIP | $196.80 \pm 53.20$ | $225.50 \pm 93.81$ | $-2.94 \pm 0.33$ | $-6.43 \pm 2.73$ | $51.70 \pm 20.40$ | $4.80 \pm 1.60$ | $3.20 \pm 1.54$ | $0.49 \pm 0.26$ | $25.60 \pm 9.79$ |
| RP | $198.60 \pm 72.36$ | $249.90 \pm 107.72$ | $-3.30 \pm 0.72$ | $-6.20 \pm 2.35$ | $53.20 \pm 29.50$ | $6.90 \pm 1.64$ | $3.40 \pm 0.92$ | $0.60 \pm 0.30$ | $23.10 \pm 16.59$ |
| IRNI | $165.80 \pm 70.00$ | $236.20 \pm 120.13$ | $-3.28 \pm 0.89$ | $-6.77 \pm 1.84$ | $44.70 \pm 19.11$ | $7.20 \pm 2.09$ | $2.60 \pm 1.11$ | $0.48 \pm 0.28$ | $20.60 \pm 19.02$ |

Table 18: The mean and standard deviation for each found hyperparameter for each method on the CEXP dataset with EoR

| Parameter | batch size | epochs | learning rate | weight decay | features | layers | dimensions | step size | EoR |
|---|---|---|---|---|---|---|---|---|---|
| RNI | $172.10 \pm 66.95$ | $404.80 \pm 66.97$ | $-3.78 \pm 0.39$ | $-6.29 \pm 2.17$ | $77.00 \pm 36.91$ | $8.30 \pm 1.55$ | $1.30 \pm 0.46$ | $0.74 \pm 0.24$ | $22.50 \pm 19.99$ |
| CLIP | $193.70 \pm 64.15$ | $304.90 \pm 121.05$ | $-3.05 \pm 0.72$ | $-5.41 \pm 2.55$ | $44.90 \pm 24.97$ | $7.90 \pm 0.94$ | $2.40 \pm 1.02$ | $0.70 \pm 0.32$ | $23.80 \pm 19.20$ |
| RP | $193.70 \pm 79.71$ | $235.20 \pm 96.69$ | $-3.57 \pm 0.46$ | $-6.56 \pm 2.48$ | $68.70 \pm 40.74$ | $7.60 \pm 1.56$ | $3.20 \pm 1.33$ | $0.73 \pm 0.26$ | $25.30 \pm 13.50$ |
| IRNI | $190.40 \pm 63.72$ | $224.10 \pm 155.54$ | $-3.02 \pm 0.67$ | $-5.86 \pm 3.11$ | $46.60 \pm 21.89$ | $8.10 \pm 1.58$ | $2.40 \pm 1.20$ | $0.53 \pm 0.32$ | $16.20 \pm 14.04$ |

Table 19: The mean and standard deviation for each found hyperparameter for each method on the CSL dataset with EoR

| Parameter | batch size | epochs | learning rate | weight decay | features | layers | dimensions | step size | EoR |
|---|---|---|---|---|---|---|---|---|---|
| RNI | $164.50 \pm 105.35$ | $464.60 \pm 102.96$ | $-3.18 \pm 0.58$ | $-7.46 \pm 2.07$ | $89.70 \pm 41.53$ | $9.10 \pm 1.04$ | $1.70 \pm 1.42$ | $0.58 \pm 0.23$ | $50.50 \pm 10.13$ |
| CLIP | $190.70 \pm 73.12$ | $366.50 \pm 124.09$ | $-2.55 \pm 0.45$ | $-7.67 \pm 1.34$ | $59.10 \pm 29.38$ | $6.70 \pm 1.00$ | $3.70 \pm 1.49$ | $0.37 \pm 0.25$ | $16.20 \pm 14.53$ |
| RP | $187.40 \pm 75.99$ | $319.70 \pm 92.42$ | $-2.39 \pm 0.42$ | $-6.95 \pm 2.11$ | $70.80 \pm 23.32$ | $7.20 \pm 0.98$ | $3.20 \pm 1.40$ | $0.54 \pm 0.28$ | $11.50 \pm 11.60$ |
| IRNI | $193.40 \pm 54.69$ | $217.40 \pm 74.28$ | $-2.63 \pm 0.45$ | $-6.22 \pm 3.23$ | $37.40 \pm 14.84$ | $6.20 \pm 1.94$ | $3.90 \pm 1.04$ | $0.43 \pm 0.32$ | $17.10 \pm 11.57$ |

## F   Trainability

Here we add an evaluation of the data from training all the models. More specifically, we consider the trainability of all the methods on the evaluated datasets. One aspect of trainability is: How easily can the model's hyperparameters be optimized? This question can be answered with BHO by evaluating how many hyperparameter optimization steps are necessary to reach a satisfactory performance. Alternatively, when comparing two models, the question can be answered by which model needs fewer steps to achieve greater performance. Considering our main experiment, we visualize the best-found performance at each BHO evaluation for each dataset and model. For each dataset and model, we have all BHO evaluations. From this, we compute each seed's best-found performance after each BHO step. We then average these seeds and visualize the BHO steps against the AUROC performance found (up to this point) in figures 4 and 5. We follow the nomenclature of table 1 in describing the datasets and the models. Note that the visualized performances are biased as we consider the maximum performance over multiple evaluations. Also, the best-found performances are not equal to the performance in table 1 as these figures use the validation set performances of the Monte Carlo cross-validation.

A clear difference is notable between the synthetic datasets TRI, TRIX, EXP, CEXP, and CSL and the practical datasets PROTEINS, MUTAG, and NCI1. Each model converges independently on the practical datasets to its optimal performance, making comparisons between the models more difficult. On the synthetic datasets, this comparison is more straightforward, as the models mostly converge to the same optimum. Notably, RNI converges slower than all other models. Additionally, considering the other models, a rough order of trainability could be surmised to be: (1) IRNI, (2) CLIP, (3) ORNI, (4) RNI, however, this order is less significant than RNIs poor trainability. This order would coincide with the amount of randomness in each model. If we consider for each model the size of the IR tree on any given graph, then the order of size would be the same. IRNI requires the smallest amount of individualizations to reach completely distinguished graphs. ORNI and RNI have maximal trees since they ignore any color information present during the IR tree construction. CLIP's first step is the same as IRNI, and then it can be roughly compared to ORNI and RNI, so its tree size is in between the others. Lastly, RNI can be considered more random than ORNI since it uses continuous random variables, which results in a random space of infinite size. The possible graphs for IRNI, CLIP, and ORNI are always finite. From this, we conclude making use of less randomness improves trainability.

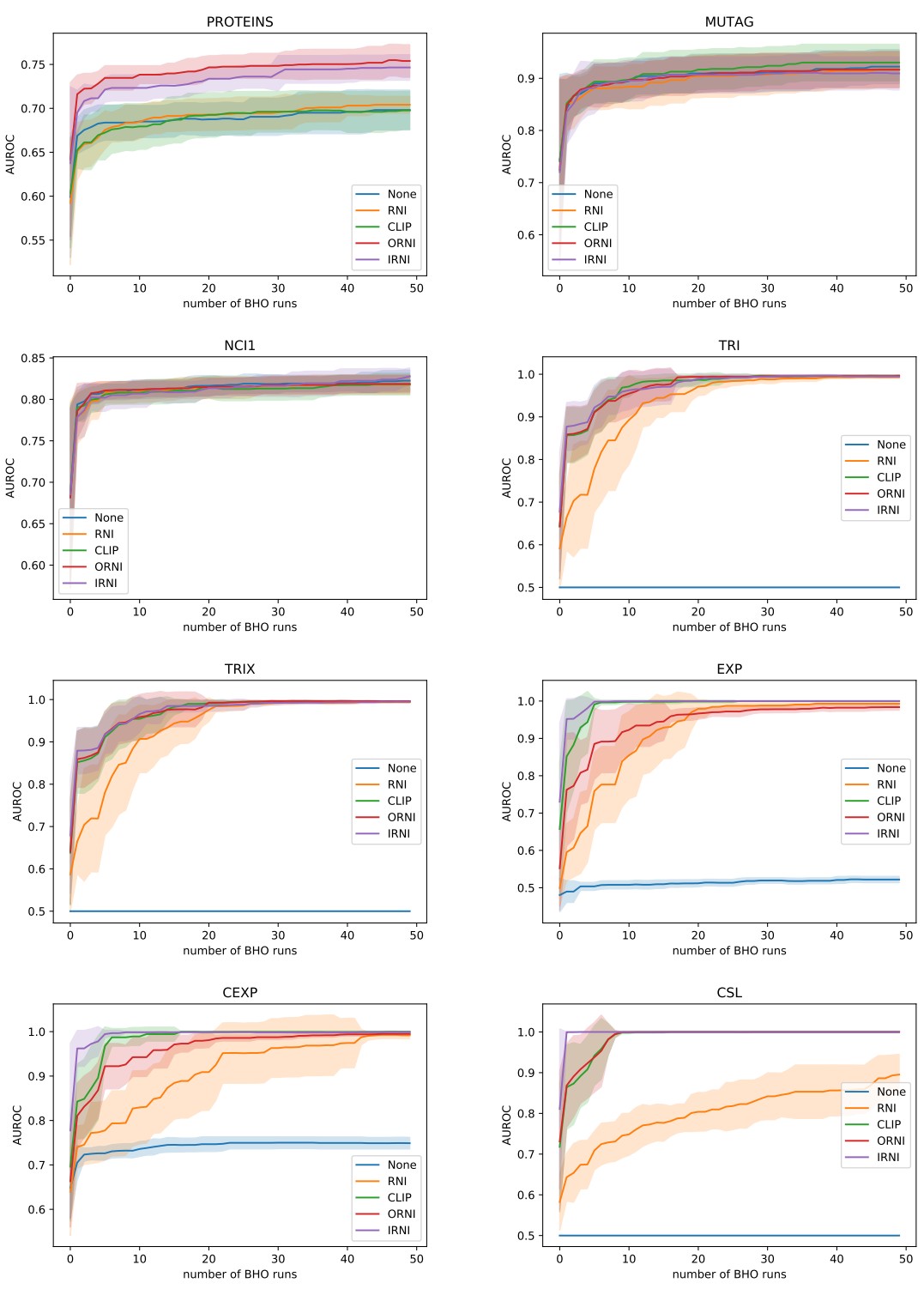

Figure 4: These plots show the mean of the best performance (y) after x BHO steps (x) over all seeds as well as the standard deviation of the BHO training from table 1 for the models without EoR.

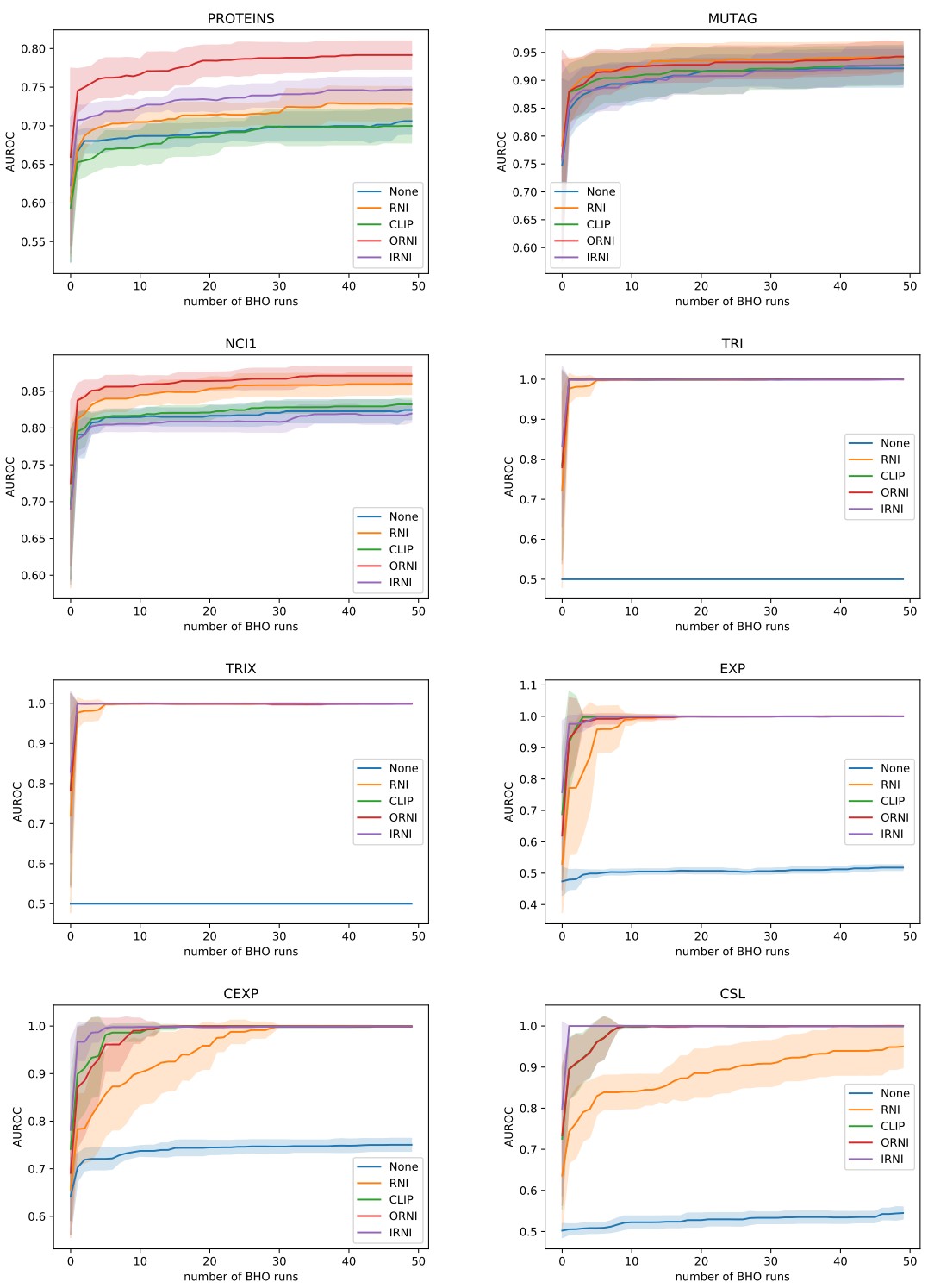

Figure 5: These plots show the mean of the best performance (y) after x BHO steps (x) over all seeds as well as the standard deviation of the BHO training from table 1 for the models with EoR.

