# OpenReview forum: "A Systematic Approach to Universal Random Features in Graph Neural Networks"
_TMLR — Accepted by TMLR_

### Review · Reviewer_SXN8 · 2023-06-06

**Summary Of Contributions:**

The existing terminology, methods, and benchmark for universal random features are highly diverse, which prevents the application of URF in practice. This paper proposes a new framework capturing all previous URF and provided comprehensive comparison for all URF. For the theoretical side, the authors formally prove the universality of all instantiations of the proposed framework IRNI under natural condition.

**Audience:**

Yes

**Claims And Evidence:**

Yes

**Requested Changes:**

1.	Provide more details on the trainability of URF.

2.	Highlight the main contribution appropriately.

**Strengths And Weaknesses:**

Strengths

1.	This paper is well-organized and well-written. The contributions are clearly illustrated in Introduction, and backgrounds on several crucial concepts are introduced.

2.	The theoretical contributions look good to me. The authors reveal the relationship between IR algorithms and MPNNs using IRNI.

Weaknesses

1.	The authors mention no systematic comparison among these URF and the trainability, generalizability, and expressivity is unclear. What’s the trainability? How do you measure the trainability?
2.	It would be better to only highlight theoretical contribution or comprehensive benchmark. The current version is insufficient to systematically compare all URF, particular in in terms of trainability and generalizability. The experiments are only conducted in GIN.
v

---

> ### Author Response · Authors · 2023-06-12
> **Author Response to Reviewer SXN8**
>
> Thank you for your encouraging review.
>
> We do indeed indicate in the introduction that trainability was previously reported as a significant issue (for RNI) and that no systematic comparison of the URF is available. Thus it is unclear how their trainability compares to one another. We also observed this for RNI in our initial experiments. For this reason, we set up the BHO training approach for a fairer comparison, hoping it would highlight this issue with RNI since hyperparameter selection appeared to be the main issue.
>
> The crucial point is that with our BHO training approach, we could not observe RNI having more trouble than other methods. None of the methods had any trainability issues in the end.
>
> For comparing trainability, we could compare the top performance after x BHO optimization steps for the various methods. However, because trainability is not an issue anymore, this may not yield interesting or conclusive results.
>
> Does this clarify the issue?

---

> ### Author Response · Authors · 2023-06-29
> **Author Response to Reviewer SXN8**
>
> Thank you again for your review. We have uploaded a revision, including changes regarding your review and requested changes.
> - We added an evaluation of trainability as Appendix F Trainability. We found a somewhat interesting result. According to this evaluation, RNI is less trainable, and trainability inversely correlates with the amount of randomness introduced by a URF. Thank you for this comment.
> - Regarding generalizability, TRI and TRIX are an experiment focussed on generalizability. Specifically for TRI, the training set and test set distributions are the same, but for TRIX, they are different. TRIX uses the same training set as TRI, but its test set comprises much larger 3-regular graphs with a different triangle distribution and general connectivity due to their regularity.

---

### Review · Reviewer_kbus · 2023-06-08

**Summary Of Contributions:**

This paper introduces Individualization Refinement Node Initialization (IRNI), a unified framework for randomized node initialization algorithms for GNNs. It is based on the Individualization-Refinement algorithm.
Existing universal random features (URF), including Relational Pooling (RP), Random Node Initialization (RNI), and Colored Local Iterative Procedure (CLIP), fall into this framework. By using this framework, this paper proposed a new URF systematically.
On the theoretical side, this paper proves universal approximation theorems in the following scenarios:
- Universality of the IRNI whose refinement is coarser than the color refinement.
- Universality for invariant functions on 3-connected planner graphs.
- Universality of a depth-fixed IRNI with the color refinement for the invariant functions on fixed-sized graphs (excluding an exponentially small number of graphs).

Empirically, this paper evaluates several URFs within the IRNI framework, whose hyperparameters are optimized by Bayesian Hyperparameter Optimization on five artificial datasets and two real datasets.

**Audience:**

Yes

**Broader Impact Concerns:**

No concerns about broader impacts.

**Claims And Evidence:**

Yes

**Requested Changes:**

P.1
- Section 1, Paragraph 2: enhance -> enhances

P.2
- Section 1, Paragraph 7 (Paragraph titled *Why individualization refinement*): This paragraph explains why IR is used. However, since there is no explanation of what IR, readers may think this paragraph is abrupt. Therefore, I suggest the description of IR before this paragraph.

P.5
- Section 3.1, Paragraph 4: The reference McKay & Piperno has a form different from the other references.
- Section 3.2, Paragraph 1: [...] colors $i$, $j$ the number of [...] -> [...] $i$, $j$, [...] (add a comma)

P.6
- Section 3.4, Paragraph 2:
  - $\boldsymbol{x}_v \leftarrow \boldsymbol{x}_v \circ (r_1, \ldots, r_d)$. I think $\circ$ usually denotes function composition. Since this usage of $\circ$ is unusual, I suggest explaining its definition.
  - [...] by assigning it a one-hot encoding of natural numbers. -> [...] by assigning a one-hot encoding of natural numbers to it.
  - $x_v$ -> $\boldsymbol{x_v}$
- Section 4.1, Paragraph 3:
  - Write the definition of $V^{\ast}$.
  - Write explicitly that $\nu \in V^{\ast}$.
  - I think $\mathrm{Ref}(G, \pi, \nu)^{-1} (v) = \{v\}$ is a little difficult to interpret. Since the value range of $\mathrm{Ref}(G, \pi, \nu)$ is $\{1, \ldots, k\}$, $v$ is incompatible as an argument of $\mathrm{Ref}(G, \pi, \nu)^{-1}$.

P.7
- Section 4.1, Lemma 1:
  - Since $\mathrm{Aut}(G, \pi)$ is only used in this lemma, the description would be more straightforward if we do not introduce the notation $\mathrm{Aut}(G, \pi)$ (Nevertheless, I think it is better to define the automorphism of a graph.)
  - Superscript $\varphi$ is undefined (e.g., $(G, \pi)^{\varphi}$)
  - $(G, \pi)^{\varphi} = [...] = G$: there is a notational inconsistency regarding whether we should include the coloring $\pi$ as a component of the graph $G$ or not.
- Section 4.1, Lemma 2: The concept of isomorphism-invariant is undefined.

P.8
- Section 4.2, Paragraph 3: *IRNI depends on the random walk in the IR tree and is thus a URF [...].*: URF, by definition (P1, Section 1, Paragraph 2), makes MPNN a universal function approximator. However, since it is not shown that IRNI is a universal approximator before this sentence, it may not be appropriate to call IRNI a URF, at least at this point.
- Section 4.3, Paragraph 1: *[...] The use of repeated random IR walks has recently been proven to be a near-optimal traversal strategy of IR trees*: I suggest explaining what you mean by near-optimal here. Also, I want some references supporting this claim.

P.9, Section 4.3, Theorem 5
- *Individualizing 3 vertices on a common face followed by color refinement suffices to make the coloring discrete.*: Is it correct to understand that the choice of face and vertices is arbitrary? In other words, is it sufficient to select *any* face in the planner graph and individualize *any* 3 vertices on the face suffices to make the color refinement discrete?

P.9--10, Theorem 4--6
- The concept of $(\delta, \varepsilon)$-approximation is not mathematically defined.

P.10, Section 4.3, Theorem 6
- $f$ is assumed to be invariant. Does it imply that the domain $\mathcal{G}'\_{n}$ is invariant under the action of the graph automorphism? Also, is it assumed that any graph in $\mathcal{G}'\_{n}$ originates has size $n$, that is, $\mathcal{G}'\_{n} \subset \mathcal{G}\_{n}$?

P.17, Appendix E
- I suggest describing the implementation details of models used in the numerical experiments, specifically, the libraries used to implement the models, Bayesian hyperparameter optimization, and evaluation code.

**Strengths And Weaknesses:**

**Strengths**

This paper provides a framework that enables us to treat existing URFs in a unified manner. It offers the following advantages:
- It provides a systematic method of making new URFs.
- It provides a unified approach for proving universality approximation theorems for randomized feature initialization.
- It provides the parameterization of URFs (by choice of refinement and selection algorithms), enabling us to evaluate these parameters' impact on empirical prediction performances.

**Weaknesses**

- There is room for improvement in writing. Specifically, I would suggest the following improvements (see Requested Changes for details):
  - Write the basics of IR (in a self-contained manner, if possible.)
  - Provide definitions for undefined mathematical terms and notations.
  - Elaborate theorems' proofs
- IRNI(CR), a new URF derived from the IRNI framework, does not outperform other URFs in numerical experiments, although its universality is shown.

**Claim and Evidence**

As far as I have checked, I found no apparent incorrect point in the proof. However, since I am unfamiliar with the Individualization-Refinement algorithm, I am not confident that I check correctly.
The paper claims to propose a new Bayesian Hyperparameter Optimization method. Indeed, as far as I know, the optimization is new in that it adds a penalty computed by the hyperparameter values. However, the experiment uses the usual nested cross-validation if I get all the information. Therefore, I have a question about whether this optimization scheme is novel.

**Audience**

The expressive power of GNNs is one of the main topics in GNN research. Universality is the most common way to evaluate models' expressive power. Several data initialization methods have been proposed to make GNNs universal (i.e., URFs). However, the relationship between them needs to be clarified. This paper provides a unified method to handle URFs, deepening our understanding. Therefore, this paper is of interest to TMLR audiences.

**Clarity**

As mentioned in the Weaknesses section, I think writing has room for improvement.
First, although this paper introduces some components of IR in Sections 4.1 and 4.2, if I get all the information, there is no explanation about how IR works and its basics. Readers without knowledge of IR may have difficulty grasping the paper's contents. I suggest adding the basic knowledge about IR (e.g., in Appendix) in a self-contained manner or providing pointers to necessary references.
Second, some mathematical terms and notations are used without definitions (see Requested Changes for details).
Finally, it would be more understandable if the proof of the theorems on IRNI universality (Theorem 4--6) were more detailed than the current one.

---

> ### Author Response · Authors · 2023-06-19
> **Author Response to Reviewer kbus**
>
> Thank you for your very detailed review. We appreciate all of the suggestions. However, we would like some clarification on a few of the suggestions and want to answer the posed questions:
>
> Write the basics of IR (in a self-contained manner, if possible.)
> We want to stress that our sentence in Section 4.1, second paragraph: "The IR paradigm is a complex machinery refined over many decades into sophisticated software libraries. We refer to Mckay & Piperno (2014) and Anders & Schweitzer (2021a) for an exhaustive description." should be interpreted as saying that a full description is well beyond the space constraint of the paper. We hoped that the reference we provided, which provides comprehensive introductions, would help the interested reader to enter the topic. Of course, we could add more details in the appendix. However, sections 3 and 4 contain the central aspects of IR required for the paper.
>
> Elaborate theorems' proofs
> We are not sure which proofs this refers to. Are any of the proofs too dense and should be expanded?
>
> Section 3.1, Paragraph 4: The reference McKay & Piperno has a form different from the other references.
> The user guide referenced here does not have a year since it is continuously updated with new versions coming out.
>
> Section 4.3, Paragraph 1: [...] Repeated random IR walks have recently been proven to be a near-optimal traversal strategy of IR trees: I suggest explaining what you mean by near-optimal here. Also, I want some references supporting this claim.
> Near-optimal refers to upper and lower worst cast bounds being roughly \Theta(\sqrt(N)) and \Theta(\sqrt(N)\log N), where N is the number of nodes of the IR-tree. Near-optimal specifically refers to the gap caused by the logarithmic factor.
>
> f is assumed to be invariant. Does it imply that the domain G'_n is invariant under the action of the graph automorphism? Also, is it assumed that any graph in G'_n originates has size n, that is, G'_n \subseteq G_n?
> f is assumed to be invariant means that on isomorphic graphs, the function gives the same value.

---

> ### Author Response · Authors · 2023-06-29
> **Author Response to Reviewer kbus**
>
> Thank you again for your very detailed review. We have uploaded a revision including all your requested changes and other points raised in your review.
> - We added an "IR in a nutshell" paragraph on page 2, which should help understand IR basics. Other than this, we also point the reader to the extensive literature on IR.
> - We made additions and changes according to most requested changes. The only exceptions are the points raised in our previous comment and the questions stated in the requested changes.

---

> > ### Comment · Reviewer_kbus · 2023-07-01
> > **Response to authors' comments**
> >
> > I thank the authors for responding to my comments and updating the draft. Here is the line-by-line response to the authors' comments.
> >
> > > We added an "IR in a nutshell" paragraph on page 2, which should help understand IR basics. Other than this, we also point the reader to the extensive literature on IR.
> >
> > I thank the authors for adding this paragraph. It is a good IR introduction and smoothly introduces readers to the next paragraph (**Why individualization refinement**.)
> >
> > > We are not sure which proofs this refers to. Are any of the proofs too dense and should be expanded?
> >
> > What I intended was similar to the [comment by Reviewer q6bL](https://openreview.net/forum?id=AXUtAIX0Fn&noteId=xttmMUSdRA). Adding more unambiguous proofs is preferable (see Additional Questions and Suggestions.)
> >
> > > Section 3.1, Paragraph 4: The reference McKay & Piperno has a form different from the other references. The user guide referenced here does not have a year since it is continuously updated with new versions coming out.
> >
> > OK, I understand. Thank you for the explanation.
> >
> > > Section 4.3, Paragraph 1: [...] Near-optimal refers to upper and lower worst cast bounds being roughly \Theta(\sqrt(N)) and \Theta(\sqrt(N)\log N), where N is the number of nodes of the IR-tree. Near-optimal specifically refers to the gap caused by the logarithmic factor.
> >
> > OK
> >
> > > f is assumed to be invariant means that on isomorphic graphs, the function gives the same value.
> >
> > Do we require that if a graph $G \in \mathcal{G}'_n$ and $G'$ is isomorphic to $G$, then $G' \in \mathcal{G}'_n$ (I think this is a natural requirement as a domain of an invariant function.)
> >
> >
> > **Additional Questions and Suggestions**
> >
> > - P.2, Section 1: section 4 -> Section 4
> > - P.4, Section 3.1: Regarding the definition of $enc(\boldsymbol{x}_v, G)$, does this signature mean that the enc function cannot use representations other than $\boldsymbol{x}_v$?
> > - P.7, Section 4.1: It would be easier to read if the notation $\mathrm{Sel}$ is introduced in the paragraph of **Cell selector**.
> > - P.8, Section 4.2: $x_i$ -> $\boldsymbol{x}_i$
> > - P.8, Section 4.2: $\boldsymbol{x}\_v \leftarrow \boldsymbol{x}\_v \circ (\boldsymbol{1}\_{w\_1 = v}, \ldots, \boldsymbol{1}\_{w_d=v})$ -> $\mathrm{concatenate}(\boldsymbol{x}\_v, \boldsymbol{1}\_{w_1 = v}, \ldots, \boldsymbol{1}\_{w\_d=v})$
> > - P.9, Section 4.3, Lemma 3: Although not explicitly stated in the statement, is it correct to understand that $f$ is a function that augments node features by $d$-$\mathrm{IRNI}(\mathrm{Ref})$, followed by MPNN? If so, this should be explicitly stated in the statement.
> > - P.10, Section 4.3, Theorem 4: *Let $\mathrm{Ref}$ be a refinement that computes colorings coarser or equal to $\mathrm{CR}$*: I want to clarify its mathematical meaning more precisely. Does it mean that for any $\nu \in V^\ast$ and $\pi$ defined by $\pi(i) = enc(\boldsymbol{x}_i, G)$, $\mathrm{Ref}(G, \pi, \nu)$ is coarser or equal to $CR(G, \pi, \nu)$?
> > - P.18, Appendix D: I want to confirm that the description in Appendix D corresponds to how the $enc$ function is defined in the experiment in Section 5?

---

> > > ### Author Response · Authors · 2023-07-03
> > > **Response to further questions and suggestions by reviewer kbus**
> > >
> > > Thank you for your further comments. We have implemented all further suggestions. We further answer posed questions below:
> > >
> > > >Do we require that if a graph $G \in \mathcal{G}'_n$ and $G'$ is isomorphic to $G$, then $G' \in \mathcal{G}'_n$ (I think this is a natural requirement as a domain of an invariant function.)
> > >
> > > Usually, graph classes are required to be isomorphism invariant. However, we do not explicitly need to state this in our theorems since we can always extend the graph class and the invariant function in the theorems to an invariant class.
> > >
> > > > P.4, Section 3.1: Regarding the definition of $enc(\boldsymbol{x}_v, G)$, does this signature mean that the enc function cannot use representations other than $\boldsymbol{x}_v$?
> > >
> > > Following our definitions, there are no representations other than $\boldsymbol{x}_v$. Thus the answer is yes in this context. However, since we only require integers for the application of dejavu, other representations could be used by defining additional encoding functions. As stated in Section 3.1 Graph and Colorings, page 5, edge representations (or edge colors) and edge directions can also be considered by transforming the graph isomorphism invariantly to an undirected not-edge colored graph following standard reductions.
> > >
> > > > P.9, Section 4.3, Lemma 3: Although not explicitly stated in the statement, is it correct to understand that $f$ is a function that augments node features by $d$-$\mathrm{IRNI}(\mathrm{Ref})$, followed by MPNN? If so, this should be explicitly stated in the statement.
> > >
> > > Since MPNNs are isomorphism-invariant, the lemma is true regardless of whether it includes the computation of the MPNN or not. Nevertheless, we adjusted the statement to include the computation of the MPNN explicitly.
> > >
> > > > P.10, Section 4.3, Theorem 4: Let $\mathrm{Ref}$ be a refinement that computes colorings coarser or equal to $\mathrm{CR}$: I want to clarify its mathematical meaning more precisely. Does it mean that for any $\nu \in V^\ast$ and $\pi$ defined by $\pi(i) = enc(\boldsymbol{x}_i, G)$, $\mathrm{Ref}(G, \pi, \nu)$ is coarser or equal to $CR(G, \pi, \nu)$?
> > >
> > > Yes, this is what is meant. We made this more explicit in the theorem statement.
> > >
> > > > P.18, Appendix D: I want to confirm that the description in Appendix D corresponds to how the $enc$ function is defined in the experiment in Section 5?
> > >
> > > Yes

---

> > > > ### Comment · Reviewer_kbus · 2023-07-04
> > > >
> > > > Thank you for your prompt reply. I am satisfied with the authors' responses.
> > > >
> > > > ----
> > > >
> > > > > Usually, graph classes are required to be isomorphism invariant. However, we do not explicitly need to state this in our theorems since we can always extend the graph class and the invariant function in the theorems to an invariant class.
> > > >
> > > > OK. My understanding is as follows: $f$ is invariant in the sense that if $G, G' \in \mathcal{G}'_n$ and $G$ and $G'$ are isomorphic, then $f(G) = f(G')$ holds. There exists $\tilde{f}: \mathcal{G}\_n \to \mathbb{R}$ such that $\tilde{f}|\_{\mathcal{G}'_n} = f$. We should define
> > > > $$
> > > > \tilde{f}(G) = \begin{cases}
> > > >   f(G') \quad \text{if there exists $G' \in \mathcal{G}'_n$ such that $G$ isomorphic to $G'$} \\\\
> > > >   0 \quad \text{otherwise}. \\\\
> > > > \end{cases}
> > > > $$
> > > > This definition is well-defined because it does not depend on the choice of $G'$. Also, the statement of Theorem 5 holds regardless of the choice of $\tilde{f}$.
> > > >
> > > > ----
> > > >
> > > > > Following our definitions, there are no representations other than $\boldsymbol{x}_v$. Thus the answer is yes in this context. However, since we only require integers for the application of dejavu, other representations could be used by defining additional encoding functions. As stated in Section 3.1 Graph and Colorings, page 5, edge representations (or edge colors) and edge directions can also be considered by transforming the graph isomorphism invariantly to an undirected not-edge colored graph following standard reductions.
> > > >
> > > > I understand. Then, is it better to denote the encoding function as $enc(v, G, X)$ to allow the encoding function to use nodes' representations other than $v$?
> > > >
> > > > ----
> > > >
> > > > > Since MPNNs are isomorphism-invariant, the lemma is true regardless of whether it includes the computation of the MPNN or not. Nevertheless, we adjusted the statement to include the computation of the MPNN explicitly.
> > > >
> > > > OK
> > > >
> > > > ----
> > > > > > P.10, Section 4.3, Theorem 4: *Let $\mathrm{Ref}$ be a refinement that computes colorings coarser or equal to $\mathrm{CR}$*: I want to clarify its mathematical meaning more precisely. Does it mean that for any $\nu \in V^\ast$ and $\pi$ defined by $\pi(i) = enc(\boldsymbol{x}_i, G)$, $\mathrm{Ref}(G, \pi, \nu)$ is coarser or equal to $CR(G, \pi, \nu)$?
> > > >
> > > > > Yes, this is what is meant. We made this more explicit in the theorem statement.
> > > >
> > > > OK
> > > >
> > > > ----
> > > > > > P.18, Appendix D: I want to confirm that the description in Appendix D corresponds to how the $enc$ function is defined in the experiment in Section 5?
> > > >
> > > > > Yes
> > > >
> > > > OK

---

> > > > > ### Author Response · Authors · 2023-07-05
> > > > > **Response to reviewer kbus**
> > > > >
> > > > > We believe there has been some miscommunication. The enc function is just a hash function that transforms arbitrary node representations $x_v \in \mathbb R^d$ to $enc(x_v, \dots)\in\mathbb N$. It is used to compute the input to dejavu, transforming X into $enc(X)=${$enc(x,\dots)|x\in X$}. What additional information is used to transform the encoding is not all that relevant. However, typically we want the function to be easy to compute, and it would thus not use any complicated properties of the graphs or other nodes. Another way of answering this is that we can work with representations in \mathbb N, and there are numerous ways to convert arbitrary node representations to ones of this type.

---

> > > > > > ### Comment · Reviewer_kbus · 2023-07-05
> > > > > >
> > > > > > Thank you for your additional explanation. I think now I understand what you meant. It is OK to keep the notation as it is.

---

### Review · Reviewer_q6bL · 2023-06-21

**Summary Of Contributions:**

The paper describes a novel GNN that uses random features and extends previous works. They show it is a universal approximator and evaluate it on several benchmarks

**Audience:**

Yes

**Claims And Evidence:**

Yes

**Requested Changes:**

- Writing should be improved, and terms should be clearly explained
- More detailed proofs

**Strengths And Weaknesses:**

Strengths:
- The work, to the best of my knowledge, is novel
-  The method is well grounded in theory, makes sense intuitively, and works on several benchmarks

Weaknesses:
- The writing should be improved. The main issue with the writing is that the author uses non-trivial terms he doesn't explain, or explain after he talks about them thus making it confusing for the reader.
Examples: (1) Universal random features (URF) never properly defined (I assume random features that make the inference a universal approximator)
(2) IR and RNI acronyms not defined
(3) Use discrete in a non-standard meaning which is confusing
(4) "Individualization refinement (IR) trees are the backtracking trees of practical graph isomorphism algorithms." isn't clear and should be explained better
- The proofs are more proof sketches. Should have more detailed proofs in the supplementary (what about proof for lemma 3?)
- ״Our crucial new insight is that ensembling over randomness significantly improves the performance of all URF, even in cases in which it had not been considered previously.״, I don't the experiments support this strong claim

---

> ### Author Response · Authors · 2023-06-29
> **Author Response to Reviewer q6bL**
>
> Thank you for your encouraging review. We have uploaded a revision addressing the mentioned weaknesses and requested changes.
> - We added further clarifications on the meaning of URF.
> - We added an IR definition on page 1 and the "IR in a nutshell" paragraph on page 2 to help explain IR.
> - We made some changes to our use of discrete. We want to mention that discrete is a standard term used in graph theory to describe colorings. We use this term in the same way. We do, however, agree that this can be confusing at times.
> - The "IR in a nutshell" paragraph should also help explain why and how IR trees are the backtracking trees of practical graph isomorphism algorithms.
> - We added a proof for Lemma 3 and expanded on the proof for Theorem 4.
> - Regarding whether EoR significantly improves performance or not. Out of all evaluations between not using EoR and using EoR, there is a total of 32, 4 instances decrease in performance slightly (IRNI(CR) on PROTEINS from 0.75->0.74, RP on MUTAG from 0.86->0.84, CLIP on CEXP from 0.99->0.97, and IRNI(CR) on CSL from 1.00->0.99). All of these decreases are insignificant. All other evaluations either do not increase significantly or do indeed increase significantly. In aggregate, we concluded that EoR increases performance in URF. We can weaken this statement by removing the significantly, but we believe this statement to be fair.

---

### Review · Reviewer_Urhj · 2023-07-06

**Summary Of Contributions:**

The paper proposed IRNI - an individualization mechanism that builds on the combination of random node features and individualization refinement algorithms. Additionally, the authors propose a training framework based on Bayesian parameter optimization for hyperparameter tuning.
The authors present a rigorous theoretical study and analysis of the proposed method, followed by a comparison of different URF methods with IRNI on several real-world and synthetic datasets.



**Audience:**

Yes

**Claims And Evidence:**

Yes

**Requested Changes:**

Please see my questions in my main review.

**Strengths And Weaknesses:**

Strengths:

* The paper addresses an open problem in an interesting topic of the use of random node features in GNNs.

* The paper (after the revisions) is easy to follow, also as a non-familiar with IR algorithm reviewer.

Weaknesses:

* Although this is not a proper weakness (but rather an observation), it is interesting that still, there is not a  clearly better method as far as random node features are concerned. I have read the discussion of the authors after the experimental section, but perhaps it would be possible to elaborate on why and when one would choose one method over the other? I think that this is an important point, in order to obtain a systematic way to work with random node features.

* In the context of random node features, the paper can benefit from discussing [1].

Questions to the authors:

* By taking random walk IR, is the 'level of randomness' decreased? at least with respect to the considered graph adjacency matrix?

* Can the authors please explain the proposed method is related to positional encoding methods that use propagation of node features [2],[3]? Specifically, with respect to the random walk IR algorithm?

* Why are the results in table 1 different between 'SOTA_{URF}' and the best of the methods {RNI,CLIP,RP, IRNI(CP)}? Are those simply results obtained in a different paper under the same settings or is there another difference?

[1] Global attention improves graph networks generalization
[2] From Local to Global: Spectral-Inspired Graph Neural Networks
[3] Graph Positional Encoding via Random Feature Propagation

---

> ### Author Response · Authors · 2023-07-10
> **Author Response to Reviewer Urhj**
>
> >I have read the discussion of the authors after the experimental section, but perhaps it would be possible to elaborate on why and when one would choose one method over the other?
>
> Our experiments seem to ultimately suggest that the answer to which URF performs best depends on the structure of the input. As such, we suggest testing the various approaches on each newly investigated dataset. We could add this statement to the paper if requested.
>
> >In the context of random node features, the paper can benefit from discussing [1].
>
> This work proposes a method as powerful as the folklore 2-WL test. Meaning this is different from the practical UNIVERSAL methods we investigate. However, we can include this as part of the discussion on related work, like data augmentations that are not universal but generally fall into the same area of expressivity.
>
> >By taking random walk IR, is the 'level of randomness' decreased? at least with respect to the considered graph adjacency matrix?
>
> We mentioned this to some degree in the appendix. Since all methods are drawing leaves from (or doing random walks in) the IR tree, the size of the tree and especially the number of leaves of the tree directly influence the number of values the random variable can take, and thus the "level of randomness." In this regard, IRNI(CR) is the most efficient since the tree is the smallest. RNI and RP are the least efficient, as their tree has the maximum possible size. CLIP is somewhere between as its first level (in the tree) follows the example of IRNI(CR), but from there, it has maximally sized subtrees.
>
> >Can the authors please explain the proposed method is related to positional encoding methods that use propagation of node features [2],[3]? Specifically, with respect to the random walk IR algorithm?
>
> If a graph has canonical positional embeddings, then the expressivity of MPNNs is already universal since the graph itself is canonical. If the positional embeddings are arbitrarily chosen, this would be equivalent to RNI. As such, the relationship depends on the specifics of the positional encoding.
>
> >Why are the results in table 1 different between 'SOTA_{URF}' and the best of the methods {RNI,CLIP,RP, IRNI(CP)}? Are those simply results obtained in a different paper under the same settings or is there another difference?
>
> The values in the SOTA rows are from other papers, while we computed all other rows with other seeds and varying implementations.

---

> > ### Comment · Reviewer_Urhj · 2023-07-10
> > **Rebuttal followup**
> >
> > I thank the authors for their responses to my questions. I have no remaining concerns or questions. I suggest that you add the clarifications made here to the final version of the manuscript.

---

### Decision · Action_Editors · 2023-07-20

**Recommendation:** Accept as is

**Comment:**

This paper introduces a novel approach which unifies existing random feature initialization methods using the individual refinement algorithm and shows the universality of Message Passing neural Networks with IRNI for several settings. They also present a Bayesian approach for hyper-parameter tuning.

The paper was appreciated by all reviewer but they noted that its writing could be improved. The authors responded very well to the questions and revised the paper to the satisfaction of the reviewers who now all agree on accepting it.

While I do not think the paper needs a minor revision, I encourage the authors to add the clarifications to the last reviewer Urhj (and positioning/references) in the camera ready version of the paper.

**Audience:**

This paper in the field of Graph Neural Networks is of interest to the audience of TMLR

**Claims And Evidence:**

The claims are supported by clear evidence as noted by the reviewers.